# Dpp dependent Hematopoietic stem cells give rise to Hh dependent blood progenitors in larval lymph gland of *Drosophila*

Nidhi Sharma Dey[1], Parvathy Ramesh[1], Mayank Chugh[1†], Sudip Mandal[2], Lolitika Mandal[1]*

[1]Developmental Genetics Laboratory, Department of Biological Sciences, Indian Institute of Science Education and Research Mohali, Mohali, India; [2]Molecular Cell and Developmental Biology Laboratory, Department of Biological Sciences, Indian Institute of Science Education and Research Mohali, Mohali, India

**Abstract** *Drosophila* hematopoiesis bears striking resemblance with that of vertebrates, both in the context of distinct phases and the signaling molecules. Even though, there has been no evidence of Hematopoietic stem cells (HSCs) in *Drosophila*, the larval lymph gland with its Hedgehog dependent progenitors served as an invertebrate model of progenitor biology. Employing lineage-tracing analyses, we have now identified *Notch* expressing HSCs in the first instar larval lymph gland. Our studies clearly establish the hierarchical relationship between *Notch* expressing HSCs and the previously described *Domeless* expressing progenitors. These HSCs require Decapentapelagic (Dpp) signal from the hematopoietic niche for their maintenance in an identical manner to vertebrate aorta-gonadal-mesonephros (AGM) HSCs. Thus, this study not only extends the conservation across these divergent taxa, but also provides a new model that can be exploited to gain better insight into the AGM related Hematopoietic stem cells (HSCs).
DOI: https://doi.org/10.7554/eLife.18295.001

*For correspondence:
lolitika@iisermohali.ac.in

Present address: †Cellular Nanoscience, Center for Plant Molecular Biology, University of Tuebingen, Tuebingen, Germany

Competing interests: The authors declare that no competing interests exist.

## Introduction

Studies in the last decade have established *Drosophila melanogaster* as a wonderful *in vivo* model to gather insights into several aspects of stem cell biology. Besides identification of signals and mechanisms involved in stem cell maintenance and differentiation, these studies have also revealed mechanisms underlying stem cell and niche interaction, essential for maintenance of healthy stem cell populations (*Losick et al., 2011*; *Pearson et al., 2009*; *Hsu et al., 2014*; *Gunage et al., 2014*; *Yuan and Yamashita, 2010*; *Lin, 2002*; *Singh et al., 2007*; *Inaba et al., 2015*).

One of the areas that have drawn much attention in recent past is the mechanism of blood cell formation or hematopoiesis in flies (*Mandal et al., 2004*, *Mandal et al., 2007*; *Krzemien et al., 2007*; *Mondal et al., 2011*; *Márkus et al., 2009*; *Makhijani et al., 2011*; *Leitão and Sucena, 2015*; *Morin-Poulard et al., 2016*). The onset of definitive hematopoiesis in *Drosophila* occurs in a defined multi-lobed larval organ, the lymph gland. The first or the primary lobe of the lymph gland houses a bunch of stem -like progenitor cells in its central region forming the medullary zone (MZ). These multi-potent progenitor cells can give rise to all blood cell lineages which populate the outer periphery of the gland, referred as the cortical zone, CZ (*Jung et al., 2005*) (*Figure 1A*). Posterior to both of these zones is the niche or the Posterior Signaling Centre (PSC), which carefully orchestrates homeostasis in the organ through an intricate regulatory network, thereby maintaining the progenitors and the differentiating hemocytes and their dainty balance (*Lebestky et al., 2003*;

**eLife digest** Blood cells perform many important roles in animals, including transporting oxygen around the body and defending against invading microbes. There are many similarities between how blood cells develop in vertebrate animals, such as humans, and in fruit flies. Therefore, researchers often use fruit flies as models to study how vertebrates make new blood cells. Fruit fly larvae produce new blood cells in the lymph gland from "progenitor" cells that show several similarities to the stem cells that make blood cells in vertebrates, known as Hematopoietic Stem Cells (or HSCs for short). The progenitor cells grow and divide and subsequently develop into mature blood cells.

The fate of the progenitor cells depend on signals produced from an adjoining group of specialized cells that serve as its niche. For example, a signal protein called Hedgehog delivered from the niche maintains the progenitor cells. On the other hand, HSCs in vertebrates rely on another signal protein called BMP to maintain them. Although the progenitor cells present in the lymph gland are believed to behave like vertebrate stem cells, it is not clear whether fruit flies also have groups of HSCs.

Dey et al. studied the fates of cells in the lymph glands of fruit fly larvae. The experiments reveal a new group of cells in the lymph gland that behave like vertebrate stem cells and give rise to progenitor cells. Since the newly discovered cells also depend on a niche for their maintenance, Dey et al. named them HSCs. Further experiments show that, like HSCs in vertebrates, the fruit fly HSCs use Dpp (the fruit fly equivalent of BMP) as a niche signal.

The next challenge is to use fruit fly HSCs as models to study how the blood develops in vertebrates, and how this is altered in individuals with blood disorders.
DOI: https://doi.org/10.7554/eLife.18295.002

Mandal et al., 2007; Mondal et al., 2011). Recent work has shown that some of the progenitors in the posterior lobes of the lymph gland home into active hematopoietic hubs in adult flies and initiate new blood cell formation and specification (Ghosh et al., 2015).

Although the progenitor cells present in the MZ are believed to be stem-like cells due to their multi-potent nature (Jung et al., 2005), intriguingly, the presence of hematopoietic stem cells (HSCs) has not yet been established. Analyses of the cells populating second and third instar larval lymph glands for properties associated with stem cells such as slow proliferation, asymmetric cell division and expression of stem cell specific markers have failed to provide any evidence for the presence of bona-fide HSCs (Krzemien et al., 2010). However, using MARCM clone based lineage analysis, Minakhina and Steward (2010) proposed the presence of HSCs in the embryonic and first instar larval lymph gland but could not provide any direct evidence for their existence (Minakhina and Steward, 2010). Thus, this popular invertebrate model of hematopoiesis till date has not been exploited to understand queries related to HSCs.

The work described here, shows the presence of a transient, hitherto unknown Notch (N) expressing cell type in the first instar larval lymph gland. This Notch positive mutipotent cell type can give rise to previously characterized Domeless expressing progenitors. Interestingly, like many other stem cells, these cells also depend on a niche for their maintenance. This prompted us to term them as the Hematopoietic stem cells (HSCs). Our genetic analyses reveal that *Drosophila* HSCs, like the mammalian AGM HSCs, use Dpp/BMP (Bone Morphogenetic Protein) as a niche signal, extending the similarities across these divergent taxa. Using this simpler yet genetically tractable model, we can now address problems regarding AGM related HSC, both at the level of developmental hematopoiesis as well as disease biology.

## Results

### The first instar lymph gland contains a unique cell type distinct from the *domeless* expressing hemocyte progenitors

While the mature lymph gland is known to have three distinct zones comprising of differentiating hemocytes (CZ), progenitors (MZ) and niche (PSC), the first instar lymph gland is believed to consist

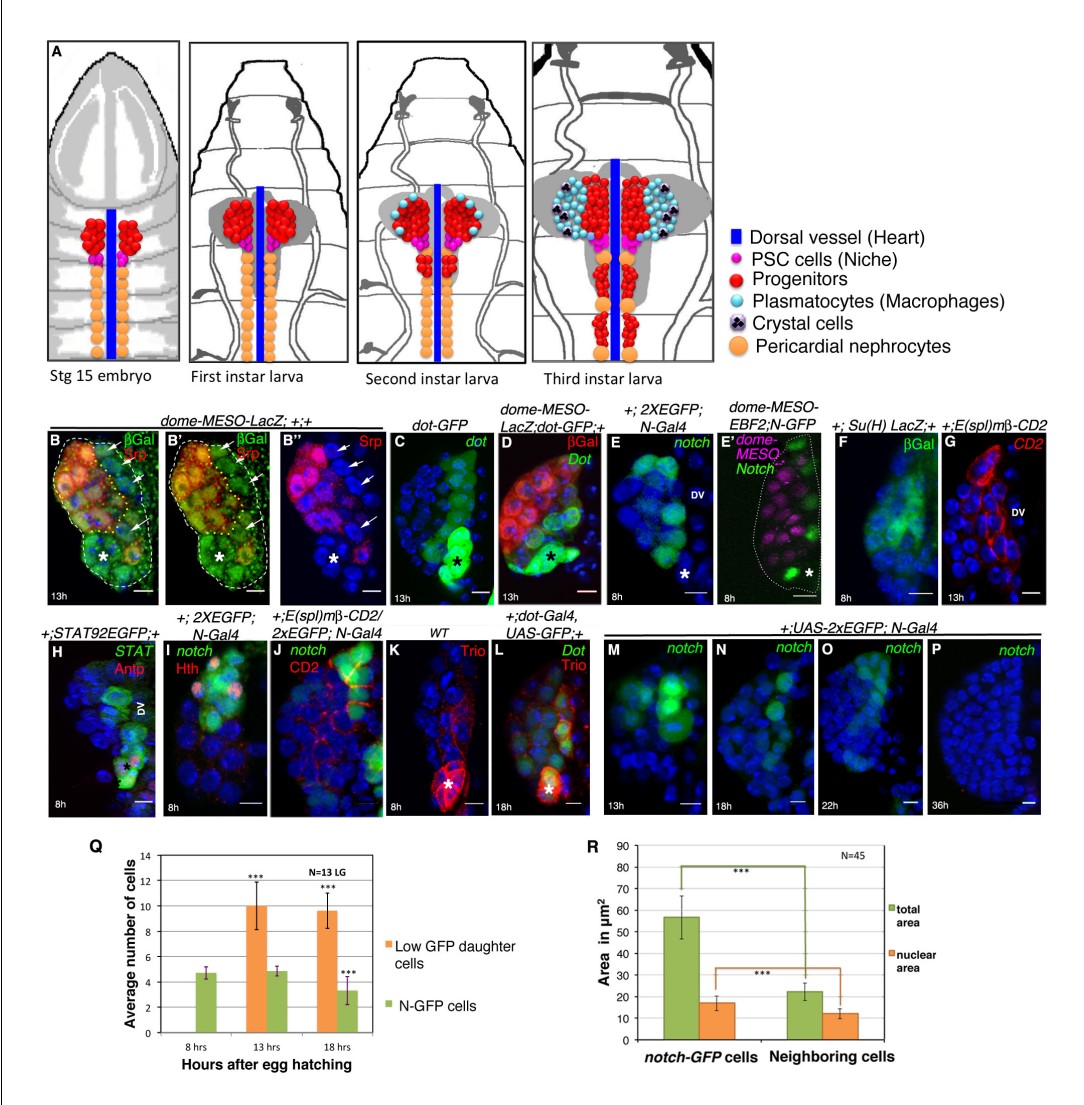

**Figure 1.** Presence of distinct cells in the first instar larval lymph glands of *Drosophila* that express several unique molecular markers. (**A**) A schematic representation of lymph gland development throughout larval life. Red: progenitors; Magenta: niche cells; Blue: dorsal vessel (DV); orange: pericardial cells; light blue: plasmatocytes; purple with black crystals: Crystal cells. (**B–B″**) shows few Serpent (Srp; green) expressing cells (arrows) that lack *domeless-MESO-LacZ* (*dome*; red) expression in a first instar lymph gland. *dome* negative cells shown in white arrows are outlined by yellow dotted lines. Asterisk marks the niche. (**C**) *dorothy-GFP* (*dot*; green) expression is higher in the PSC compared to the cells that align near the dorsal vessel, but are negative for *dome* (red) expression in (**D**). Also see *Figure 1—figure supplement 1E*. (**E–G**) Expression of *Notch* alone (N; green; [**E**]), co-staining with *domeless* expressing progenitors (magenta, E′) and its pathway components: *Suppressor* of *hairless* (Su(H); Green; **F**) and *Enhancer* of *Split* (E(spl); red; [**G**]) in the cells near the DV. (**H**) *STAT92EGFP* also expresses in these cells in addition to the PSC (red, Antp indicated by an asterisk). (**I**) A subset of *N* expressing cells (green) are positive for Homothorax, (hth; red) expression. (**J**) Overlap of *E(spl)* (red) and *N* (green; n = 10) expression. (**K–L**) indicate Trio (red) expression in the PSC (asterisks) and in cells near the dorsal vessel, which overlaps with *dot-Gal4, UAS-GFP* as evident in (**L**). (**M–P**) shows expression of *N* (green) in cells close to the DV during early first instars. Also see **E** and *Figure 1—figure supplement 1I*. This expression is hardly detectable beyond 22 hr AEH (**O–P**) (**Q**) Quantitative analysis of the number of *N* expressing cells with respect to time. Based on the fluorescence intensity estimation in (*Figure 1—source data 1*), number of *N* expressing cells and their daughter cells are 4.8 at 8 hr, 4.8 (p=0.635406062, two tailed unpaired Student's t-test) and 10 (p=2.14882E-10, two tailed unpaired Student's t-test) at 13 hr and 9.6 (p=1.01648E-11, two tailed unpaired Student's t-test) and 3.3 (p=0.000754707, two tailed unpaired Student's t-test) at 18 hr AEH respectively. (**R**) Quantitative estimation of the nuclear and total area of the *N* expressing cells with respect to neighboring cells. The total area of *N* expressing cells is 2.5 times (n = 45; p=8.50672E-29, two tailed unpaired Student's t-test) and nuclear area is about 1.4 times (n = 45; p=1.68523E-11, two tailed unpaired Student's t-test) greater than surrounding cells. Scale bar = 5 μm. Error bars=SD. Genotypes are shown on top of corresponding panels. DAPI marks the nucleus. Hours after larval hatching are as indicated in each panel. Also see *Figure 1—figure supplement 1–2*.

DOI: https://doi.org/10.7554/eLife.18295.003

The following source data and figure supplements are available for figure 1:

*Figure 1 continued on next page*

*Figure 1 continued*

**Source data 1.** Contains numerical data plotted in *Figure 1Q and R* and Total Fluorescence Intensity analyses for Q and R.
DOI: https://doi.org/10.7554/eLife.18295.006
**Figure supplement 1.** *Notch* expressing cells lack the earliest progenitor marker at first instar stage.
DOI: https://doi.org/10.7554/eLife.18295.004
**Figure supplement 2.** Arrangement of *Notch* expressing cells with respect to rest of the cells in a first instar lymph gland.
DOI: https://doi.org/10.7554/eLife.18295.005

of only the PSC and the MZ (*Figure 1A*)(*Krzemien et al., 2010*; *Jung et al., 2005*). However, our analyses of the first instar larval lymph gland revealed the presence of few Serpent positive (a pan-blood cell marker; [*Rehorn et al., 1996*]) cells that lacked the expression of *dome-MESO-lacZ (domeless)* (arrows, *Figure 1B-B''*), a bona-fide marker for prohemocytes (*Jung et al., 2005*) (*Figure 1—figure supplement 1A*). Interestingly, this cell type was aligned next to the larval heart, the dorsal vessel (DV, *Figure 1B–B''*). To further ascertain this finding, we checked the expression of two reporter fly lines: *dome-Gal4, UAS-GFP* along with *dome-MESO-LacZ* (*Krzemien et al., 2007*). In doing so, we could track these distinct cells that were negative for both of these fly reporter constructs (*Figure 1—figure supplement 1B–B''*). Together, the above results clearly establish the presence of few cells in the MZ that are distinct from the prohemocytes in the first instar lymph gland.

Next, we looked for the markers uniquely expressed in these *dome* negative cells for further analysis. Interestingly, using *dorothy-Gal4, UAS-GFP* (Dot) (*Honti et al., 2010*), a pan embryonic lymph gland marker, we were able to trace this new cell type shortly after the emergence of larvae. We found that the *dot-GFP* expression was not uniform in the newly hatched larval lymph gland. While majority of the lymph gland cells showed a very low expression of *dot* (*Figure 1—figure supplement 1C*), few cells near the dorsal vessel expressed relatively higher level of *dot*. Additionally, expression of *dot* could also be seen in the PSC (co-localization with Antennapedia, Antp expression ([*Mandal et al., 2007*; *Jung et al., 2005*]; *Figure 1—figure supplement 1D*). Intriguingly, the cells that aligned next to the dorsal vessel with relatively higher levels of *dot* (*Figure 1C*) lacked *dome-MESO-lacZ* expression (*Figure 1D* and *Figure 1—figure supplement 1E*).

These findings show the presence of a new blood cell type, adjacent to the dorsal vessel in the early first instar lymph gland, that is yet to turn on the progenitor markers. This prompted us to presume these cells to be the founder cells of the prohemocytes.

## The novel cell population is transient and expresses several molecular markers

Further exploration resulted in identification of Suppressor of Hairless (*12XSu(H)LacZ*) and Enhancer of Split *E(spl)mβ-CD2*, the transcriptional reporters of Notch (N) signaling (*Weinmaster, 1998*; *Jennings et al., 1994*; *Fre et al., 2005*) being expressed in these cells (*Figure 1F–G* and *Figure 1—figure supplement 1F–H''*). Likewise, we also found that these cells expressed STAT of conserved cytokine-activated Janus kinase/signal transducer and activator of transcription (JAK/STAT) pathway, (*Hou et al., 1996*) as well as Homothorax (Hth; *Drosophila* Meis 1 homologue (*Kurant et al., 2001*) (*Figure 1H–I*). Interestingly, both of these signals, JAK/STAT and Homothorax have been previously shown to be involved in the lymph gland development. While STAT has been studied extensively during the late larval hematopoiesis, homothorax has been shown to specify lymph gland proper at late embryonic stages (*Mandal et al., 2007*). JAK/STAT pathway has been shown to both autonomously and non autonomously maintain the quiescence state of hematopoietic progenitors, thus, preventing differentiation (*Krzemien et al., 2007*; *Mondal et al., 2011*; *Minakhina et al., 2011*).

Quite strikingly, these signals have also been known to act on Hematopoietic Stem Cells (HSCs) and are active in setting up definitive hematopoiesis in vertebrates. While Meis1, the Hth homologue in vertebrate has been unique to blood, the other signals have been recognized as key players in stem cell maintenance in many other developmental contexts (*Koch et al., 2013*; *Stine and Matunis, 2013*; *Argiropoulos et al., 2007*; *Unnisa et al., 2012*; *Varnum-Finney et al., 1998*; *Burns et al., 2005*; *Butko et al., 2016*; *Gama-Norton et al., 2015*). Therefore, the presence of these signature molecules restricted to these newly identified cell type was an encouraging indication, that we might be observing the elusive HSCs in the first instar larval lymph gland.

We also found Trio, a unique Guanine Nucleotide-Exchange factor (GEF) in these cells and in the PSC. (*Figure 1K–L*). Although Trio has not yet been associated in stem cell biology, intriguingly, - few other GEFs have been implicated in mammalian and invertebrate stem cells scenarios. For instance, GEF Vav1 is shown to regulate perivascular homing and bone marrow retention in mammalian HSCs and progenitor cells (*Sanchez-Aguilera et al., 2011*). Likewise, another GEF (gef26) adheres testis' GSC to its niche in *Drosophila* (*Wang et al., 2006*).

Having clearly identified this distinct cell type, we next wanted a specific driver to manipulate this cell population. Our effort yielded in identification of *Notch (N)-Gal4* (*Figure 1E*) as a definite driver - as it excludes both progenitors (*domeless*, magenta: *Figure 1E'*) and the niche (Antennapedia, red: *Figure 1—figure supplement 1G*). This driver was further validated by co-localization studies with *E(spl)mβ-CD2* and *12XSu(H)LacZ* (*Figure 1J* and *Figure 1—figure supplement 1H–H''*) reporter expression. Time kinetic studies with *N-Gal4, UAS-2XEGFP* revealed the presence of this distinct cell type at 8 hr after egg hatching (AEH; *Figure 1E–E'*). Interestingly, it was observed that these newly identified GFP expressing cells were bigger in size than the rest of the prohemocytes. Thereafter, by 13 hr AEH, few small cells with lower GFP expression were seen, in addition to the bigger cells with higher GFP expression (*Figure 1M*). It was observed that at about 18 hr AEH, high GFP expressing cells were much less in number, as compared to the low GFP expressing ones (*Figure 1N* and *Figure 1—figure supplement 1I*). Interestingly, beyond 22 hr AEH, they were hardly seen (*Figure 1O–P*). A quantitative analysis revealed a 7-fold reduction in the GFP fluorescence intensity in the smaller cells when compared to the bigger cells expressing high levels of *Notch* (*Figure 1—source data 1*). On counting these two cell populations at different time intervals (8, 13 and 18 hr AEH), we realized that although the number of bigger cells remains constant (4–5 cells by 13 hr AEH), there was a considerable increase in the number of smaller cells indicating a division at/around this time (*Figure 1Q*).

In addition, the *Notch* (*N*) expressing cells when compared to the surrounding smaller cells were found to be about 2.5 times and 1.4 times bigger in total area and nuclear area respectively (*Figure 1R*).

A 3D re-construction of a confocal image of *dome-MESO-LacZ* expression (*Figure 1—figure supplement 2A–A'*; *Video 1*) clearly demonstrated the existence of this previously unidentified cell type in the early larval lymph gland (*Figure 1—figure supplement 2B*), which is distinct from the progenitor population in terms of different molecular markers, count and size.

## Notch expressing cells in the first instar lymph gland undergo asynchronous division from 13 AEH

To determine the nature of proliferation of the *Notch* positive big cells, we induced transient labeling of the nucleus by H2B-YFP. This transient labeling was achieved by heat shock activation of *UAS-Histone2B-YFP*, in late 16 stage of embryonic development for an hour (*Figure 2A*). To check if all cells were labeled at time 0, in each batch, few embryos were analyzed for YFP label post 2 hr of heat shock. In all cases, the embryonic lymph gland (lg) was uniformly labeled (lg, *Figure 2B–B''''*). For determining the retention of labeling, rest of the embryos were reared till first instar larval stage and dissected at 12, 13 and 18 hr AEH.

Interestingly, we found that at 12 hr AEH while all the cells of the lymph gland had lost or had low levels of YFP expression, 4–5 cells near the dorsal vessel seemed to have higher levels of YFP (*Figure 2C*). We observed a further dilution at 13–18 hr AEH, when only two to three cells near the dorsal vessel retained YFP expression (*Figure 2D–F*). These observations clearly indicated that the cells with higher YFP expression must have divided by this time point.

To monitor the cell cycle status of the transient label retaining cells, we employed two

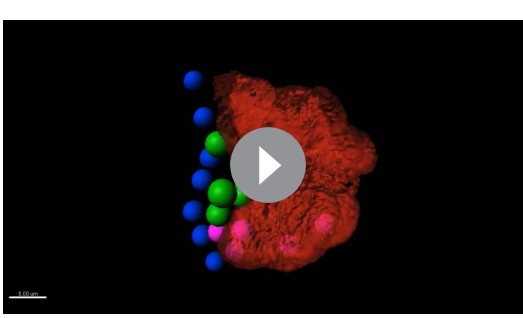

**Video 1.** Position of *Notch* expressing cells in the first instar lymph gland with respect to progenitors (Red), PSC (Magenta) and the dorsal vessel (Blue). Also see *Figure 1—figure supplement 2*.
DOI: https://doi.org/10.7554/eLife.18295.007

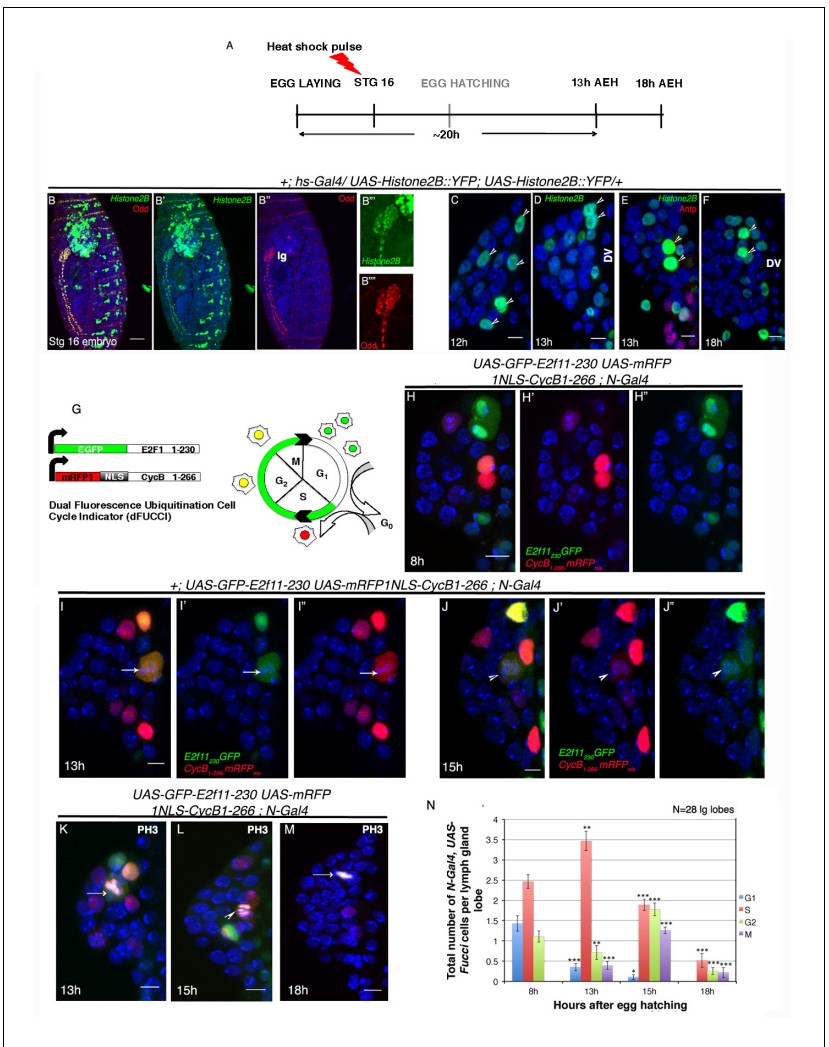

**Figure 2.** Cell cycle analysis of label retaining *Notch* expressing cells in first instar lymph gland of *Drosophila*. (A–F) Transient activation of *UAS-Histone2B::YFP* by *hs-Gal4* following scheme in (A) shows labeling of all cells of the embryonic lg at 2 hr post hs ([B–B"'']; red; odd; n = 10). (C–F) Subsequent analyses of the rest of the embryos at 12 hr ([C]; n = 18), 13 hr ([D–E]; n = 20) and 18 hr (F; n = 10) AEH, reveals few high YFP cells near DV retaining label (shown by arrow-heads). (G) The dual *Fucci* system in fly. Cells fluoresce green in G1, red in S and yellow in G2-M phases of cell cycle. (H–H") At 8 hr AEH, *N* expressing cells are predominantly in G1 (only green) and S (only red) phases of cell cycle, whereas by 13 hr (I–I"), majority of these cells can be seen piling up in S phase (p=0.00121289, n = 28, two tailed unpaired Student's t-test). Mitosis (cytoplasmic yellow, arrow, I) in these cells can also be detected from 13 hr onwards (p=0.00027463, n = 28, two tailed unpaired Student's t-test). (J–J") show cell cycle status of *N* expressing cells at 15 hr, where they can now progress into G2 (yellow (J), p=1.79435E-05, n = 28, two tailed unpaired Student's t-test) and M phases of mitosis (yellow, arrow head, p=8.49479E-09, two tailed unpaired Student's t-test). (K–M) PH3 labeling in the lymph gland 13 hr and 15 hr AEH (M) No *Fucci* signal can be detected at 18 hr AEH (S phase, p=4.17476E-08; G2 phase, p=1.26203E-10; M phase, 2.30293E-07; n = 28; two tailed unpaired Student's t-test). (N) Quantitative analysis of results from H–M. Average number of cells in each phase of cell cycle was counted and plotted. DV= Dorsal Vessel. Scale bar = 5 μm. Error bars=SE. Genotypes are shown on top of corresponding panels. DAPI marks the nucleus. Hours after larval hatching are as indicated in each panel.

DOI: https://doi.org/10.7554/eLife.18295.008

The following source data and figure supplements are available for figure 2:

**Source data 1.** Contains numerical data plotted in *Figure 2N* and *Figure 2—figure supplement 1E*.
DOI: https://doi.org/10.7554/eLife.18295.011

**Figure supplement 1.** Cell cycle analyses of *Notch* expressing cells.
DOI: https://doi.org/10.7554/eLife.18295.009

*Figure 2 continued*

**Figure supplement 2.** Absence of apoptosis in *Notch* expressing cells in first instar lymph gland of *Drosophila*.
DOI: https://doi.org/10.7554/eLife.18295.010

independent *UAS-Fucci* systems. We first used a dual *Fucci* construct (*Zielke et al., 2014*) that has two different probes that can clearly distinguish between G1, S and G2 phases of cell cycle. The first probe uses E2F moiety fused to GFP that is efficiently degraded by Cdt2 during S phase of cell cycle and is thereby able to identify G1, G2 and M phases. The second probe uses a CycB moiety fused to mRFP and is susceptible to degradation by APC/C during G1 phase, thereby clearly identifying cells in S, G2, and M phase. Thus essentially, cells from anaphase to the G1/S transition are green, S phase are red while cells in G2 and early mitosis are yellow (*Figure 2G*). To further demarcate G2 and M phases of cell cycle, we analysed nuclear staining (DAPI) as well as M phase marker, Phospho-histone 3.

On driving this construct with *N-Gal4*, we found that at 8 hr AEH, although majority of the cells were in S phase, few of them were also in G1 and G2 phases (*Figure 2H–H", N*). With time, as the G1 cells moved into S phase, we observed an increase in the number of S phase cells by 13 hr AEH. At this time point we could see a reduction in G2 cell number indicating a shift to M phase (arrow, *Figure 2I–I", N*). These results coupled with PH3 labeling (*Figure 2K,N*) confirmed that around 13 hr AEH the *Notch* expressing cells started dividing. This shift to M phase became more prominent by 15 hr, when more of these cells entered division. Additionally, we could also see a reduction in number of cells in S phase and a concomitant surge in G2 (*Figure 2J–J", L, N*).

In majority of the cases we did not detect any signal at 18 hr AEH (*Figure 2M,N*) indicating a possibility of complete division achieved by the *Notch* expressing cells.

The second construct harbored a *UAS-S/G2/M-green fluorescent ubiquitination-based cell cycle indicator* (*Fucci*, [*Sakaue-Sawano et al., 2008*; *Nakajima et al., 2011*), which shows fluorescence during the S/G2/M phases of the cell cycle only (*Figure 2—figure supplement 1A*). Nuclear localization of GFP indicates S/G2 whereas, cytoplasmic GFP reflects M phase. Since the newly identified cells were positive for *E(spl)mβ-CD2*, a fly stock *E(spl)mβ-CD2 N-Gal4* was generated and crossed with *UAS-Fucci*. In the progeny, nuclear localization of the *Fucci* was observed primarily till 13 hr AEH, indicating that the *Notch* positive cells were predominantly in S/G2 (*Figure 2—figure supplement 1B–C*). We could however, also detect few cells with cytoplasmic localization of *Fucci* (from 13 hr AEH), which signify their entry into mitosis (arrowhead, *Figure 2—figure supplement 1B-C*). This became more obvious with the increase in number of *E(spl)mβ-CD2* positive cells around 13–18 hr AEH indicating the emergence of progenitors at this time point (*Figure 2—figure supplement 1C–D*).

Cytoplasmic localization of *Fucci* was further validated by Phospho-histone 3 (*Figure 2—figure supplement 1F–G*) indicating that the transient label retaining *Notch* positive cells undergo asynchronous division to seed off probable progenitors during this time. Quantitative analyses of this data revealed that indeed maximum cell division takes place between 13–18 hr AEH wherein majorly all *Fucci* labeled *Notch* positive cells enter- into mitosis (*Figure 2—figure supplement 1E*).

In majority of the cases with both the *Fucci* constructs, we did not detect any signal at 18 hr AEH (*Figure 2M* and *Figure 2—figure supplement 1H*), indicating a possibility of complete division achieved by the *Notch* expressing cells.

In this context, it was interesting to note that the number of *Notch* positive cells in late embryonic stages matches with that of 8 hr AEH (Compare *Figure 1E* with Figure 4A-B'). Moreover, the label induced in late embryogenesis was retained in 4–5 cells till 12 hr AEH (*Figure 2C*). Both the above observations also indicated that possibly no cell division has occurred till 8–12 hr. This along with detailed *Fucci* analyses clearly showed that the *Notch* expressing cells undergo one cell division post-hatching and the asynchronous population completes this round within 13–18 hr AEH.

Since we observed a reduction in the number of Notch positive cells post 13 hr AEH followed by their absence beyond 22 hr AEH, we investigated if these cells undergo cell death eventually. For this purpose we activated a sensor of apoptosis, *UAS-Apoliner5* (*Bardet et al., 2008*) by *N-Gal4*. This sensor fly line harbors two fused fluorescent proteins that on caspase activity get differentially localized. While in non-apoptotic cells, the two fluorescent proteins mRFP and NLS-EGFP reside at the membrane, during apoptosis in the presence of active caspases, the NLS-EGFP gets

accumulated in the nucleus. Interestingly, we could not detect any nuclear localized EGFP signal in the *Notch* expressing cells during this time line indicating that none of these cells undergo cell death (*Figure 2—figure supplement 2A–D"*).

## The *Notch* expressing cells in the first instar lymph gland are multi-potent stem cells

Previous studies using the *G-TRACE* construct have described *dome* expressing progenitors as multi-potent cells (*Jung et al., 2005*). G-TRACE (*Evans et al., 2009*) is a dual color labeling technique, where in initiation of reporter expression (GFP or RFP) is *Gal4* dependent while its maintenance is not. Thus, activation of *G-TRACE* construct by a *Gal4* in the past is perpetuated as GFP expression in the lineage, whereas RFP marks the live expression of the gene tagged with *Gal4*.

Activation of the construct post 18 hr AEH by *dome-Gal4* (*Figure 3—figure supplement 1A–C"*), resulted in labeling of all cells of the lymph gland (*Figure 3—figure supplement 1D–D'''*, Hindsight, Hnt: Crystal cells (*Terriente-Felix et al., 2013*) and *Figure 3—figure supplement 1E–E'''*, P1: Plasmatocytes (*Kurucz et al., 2007*) indicating that they are multipotent in nature.

We were then intrigued to know the fate of the *Notch* positive big cells observed in this study. Using a temperature sensitive *Gal80* allele, we next activated the *G-TRACE* construct by N-Gal4 (*Figure 3A*) for a short period, from stage 16 of embryogenesis to 18 hr AEH, a timeline that overlaps with the existence of these *Notch* expressing cells (*Figure 3B*). To our surprise, this transient activation resulted in labeling of majority of the cells of the third instar larval lymph gland (*Figure 3C–C''*) including the differentiated cells marked by Hnt (*Figure 3D–D''*) and P1 (*Figure 3E–E''*). Since, the lymph gland consisted of both *Notch* positive and *Notch* negative cells during the period of *G-TRACE* activation (*Figure 1E,M–O* and *Figure 3A*) we speculated that the unlabeled cells are the clonal expansion of the latter.

We, next, attempted to induce clones at a single cell level to probe whether a single labeled HSC can expand and differentiate into different cells in this lineage. To induce clones at a low frequency, we lineage traced during 14–18 hr post hatching (when there are 2–3 *Notch* expressing cells in the first instar lymph gland instead of 5; *Figure 3F*). This resulted in restrictive lineage tracing of one (*Figure 3H*) or two (*Figure 3G*) HSCs. We found that their clonal population spanned from inner core to the outer periphery of the lymph gland thus encompassing both progenitors (MZ cells) and differentiated hemocytes (CZ) (*Figure 3G–H*; P1: *Figure 3I* and Hnt: *Figure 3J*). This clearly established the multi-potent nature of the *Notch* positive cells.

However, activation of the construct beyond 30 hr AEH (*Figure 3-figure supplement 2A-C*), did not result in labeling of all cells. Rather only the crystal cells were labeled by both GFP and RFP (also seen by Hnt). This crystal cell specific restriction of GFP and RFP expression can be explained by requirement of Notch for maturation, maintenance and survival of this cell type (*Lebestky et al., 2003*; *Mukherjee et al., 2011*; *Terriente-Felix et al., 2013*; *Duvic et al., 2002*; *Ferguson and Martinez-Agosto, 2014*).

Put together, these results undoubtedly validate that the lineage of the distinct *Notch* positive cells of the early lymph gland encompasses not only progenitors but also mature differentiated blood cells.

## *Notch* expressing multi-potent cells are the founder cell for progenitors of the larval lymph gland

Our investigation so far has established that the newly identified *Notch* positive cells are indeed multi-potent. Previous studies (*Jung et al., 2005*) have demonstrated that *dome* expressing progenitor cells are also multi-potent. Therefore, the next important question was to determine the hierarchical relationship between *Notch* positive cells and *dome* positive cells.

To address this issue we employed expression studies, detailed characterization of *Notch* lineage-traced cells at different developmental time points as well as genetic ablation of the Notch cells. Our investigation revealed that by late 16 stage of embryogenesis, expression of *Notch* becomes restricted to 4–5 cells in the embryonic lymph gland visualized by Odd skipped (Odd) expression (arrows, N-Gal4, 2XEGFP; *Figure 4A–A'*: lateral view and *Figure 4B–B'*: dorsal view). Odd is a transcription factor and is known to be the earliest marker of lymph gland (*Ward and Coulter, 2000*; *Mandal et al., 2004*). We were then intrigued to know the expression of *dome* in embryonic lymph

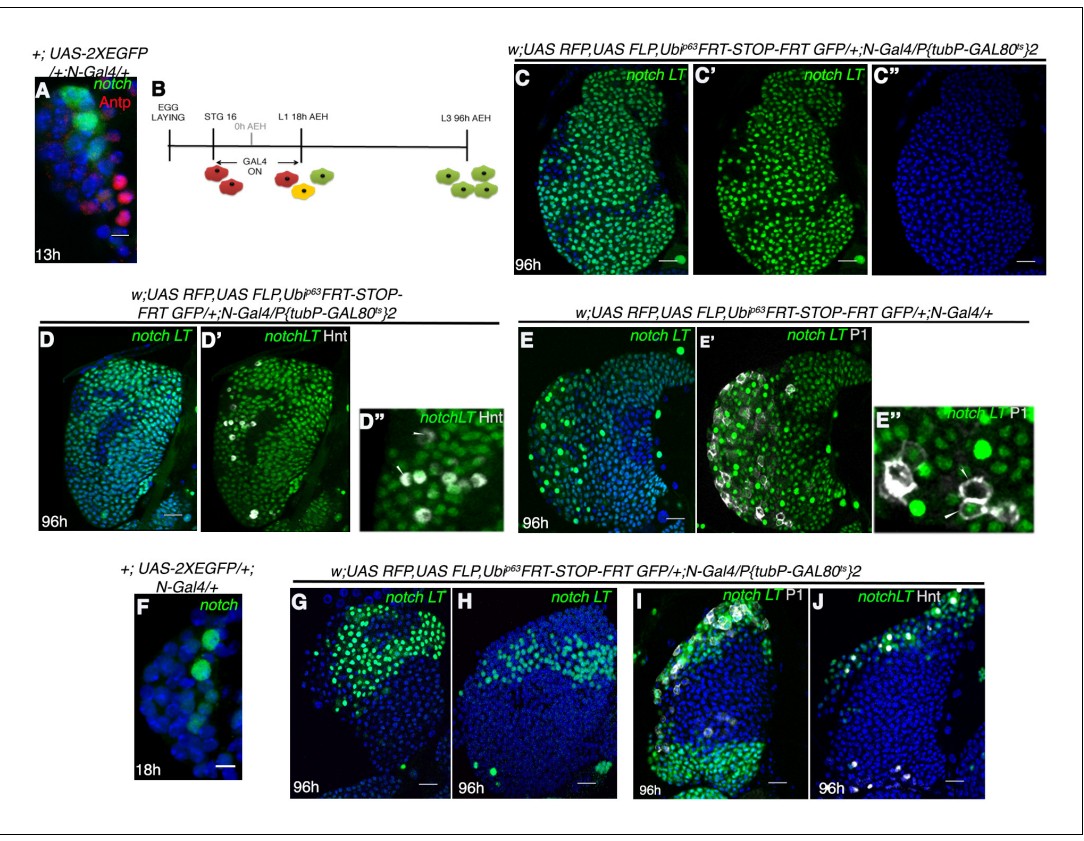

**Figure 3.** *Notch* expressing cells in first instar lymph gland of *Drosophila* are multipotent. (**A**) *N-GFP* expression in a first instar lymph gland. N expression is detectable only till late first instar (also see *Figure 1*[**E, M–O**]). (**B–E"**) shows that activation of *G-TRACE* with *N Gal4*, in a short window (following scheme in **B**) results in labeling of majority of the cells of third instar lymph gland. *N* lineage traced cells (green; [**C–E"**]; n = 15) include both crystal cells ([**D'-D"**], hnt, grey; n = 10) and plasmatocytes ([**E'–E"**], P1, grey; n = 10). (**F**) N expression is reduced to 2 cells by about 18 hr AEH (n = 7). (**G–H**) show that restrictive lineage tracing of one (**H**; n = 6) or two (**G**; n = 8) HSCs result in a clonal propagation, which spans through the entire horizontal length of the gland. (**I–J**) Each of these cells is equipotent in giving rise to both plasmatocytes ([**I**], P1, grey, n = 9) and crystal cells ([**J**], hnt, grey; n = 9). Scale bar = 5 μm for *Figure 3A and F*, for rest is 20 μm Genotypes are shown on top of corresponding panels. DAPI marks the nucleus. Hours after larval hatching are as indicated in each panel. Also see *Figure 3—figure supplement 1–2*.

DOI: https://doi.org/10.7554/eLife.18295.012

The following figure supplements are available for figure 3:

**Figure supplement 1.** Fate map of the progenitors arising from *Notch* expressing HSCs.

DOI: https://doi.org/10.7554/eLife.18295.013

**Figure supplement 2.** *Notch* requirement in crystal cell maintenance at a later stage in a third instar lymph gland.

DOI: https://doi.org/10.7554/eLife.18295.014

gland. Interestingly, while mesodermal *dome* expression can be seen in the gut (arrow; *dome-MESO-LacZ*; *Figure 4C*: dorsal view and *Figure 4E*: dorso-lateral view), we could not detect any expression of *dome* in the embryonic lymph gland (*Figure 4D–D'*: dorsal view and *Figure 4F–F'*: dorso-lateral view). Likewise, (*Mandal et al., 2004*) expression of *tepIV* (*tepIV-Gal4, 2xEGFP*), a JAK/STAT responsive thioester containing protein 4-coding gene (*Irving et al., 2005*), was also absent from the lymph gland in a similarly staged embryo (*Figure 4G–H'*: dorsal view and *Figure 4I–J'*: lateral view). It is thus evident from these results that although Notch signaling is active in the late embryonic lymph gland, JAK/STAT pathway has not yet been elicited.

To investigate whether these Notch cells, encountered in embryos, eventually turn on *dome*, we adopted the following strategy: a fly line harboring *dome-MESO-EBF2* (*Evans et al., 2014*) was brought in the background of *N-Gal4* and then lineage traced following the scheme shown in

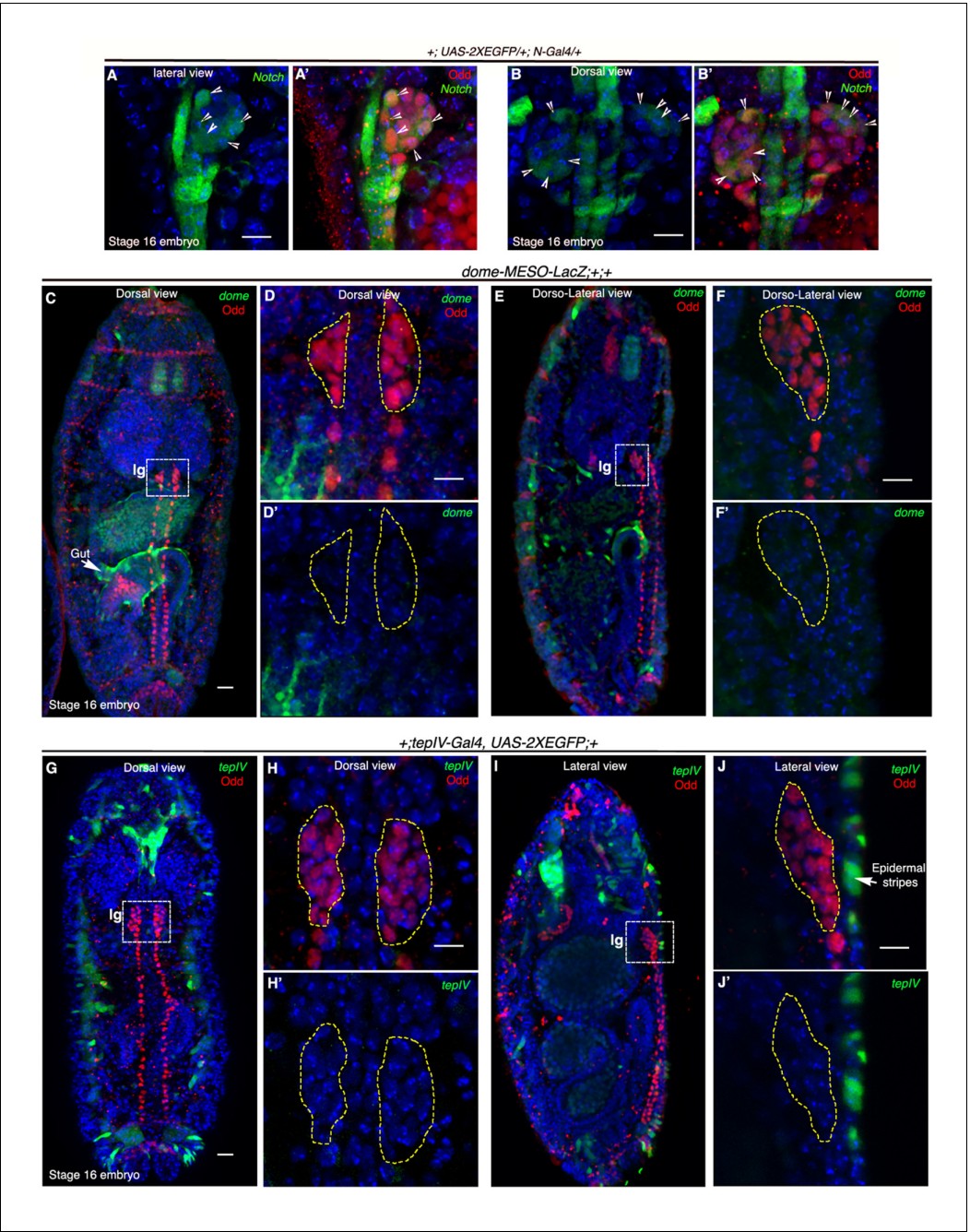

**Figure 4.** *Notch* is expressed earlier than *Domeless* in the embryonic lymph gland. (A–B') Restriction of *N-Gal4, UAS-2XEGFP* expression in 4–5 cells (arrow heads) of late 16-stage embryonic lymph gland (Odd, red). A–A' shows lateral and **B–B'** depict dorsal views (n = 15). (C–F') Mesodermal specific *domeless* expression (*domeless-MESO-LacZ*; green) is absent in embryonic lymph gland (lg, dotted white box; Odd; red) but present in the gut (arrow; n = 8). -C–D'- show dorsal while E–F' show dorso-lateral view of embryonic lymph gland. -D–D'- are insets into the dorsal view of the embryonic lymph gland in C- whereas F–F' are insets into the dorso-lateral view in E. (G–J') TepIV, a downstream responder of JAK/STAT pathway (*tepIV-Gal4, UAS-2xEGFP*; green) is also absent in the embryonic lymph gland (dotted white box; Odd; red) but present in epidermal stripes (arrows in J; n = 13). **G–H'** show dorsal while **I–J'** show lateral view of embryonic lymph gland. **H–H'** are insets into the dorsal view of the embryonic lymph gland in (G; dotted white box) whereas **J–J'** are insets into the lateral view in (I; dotted white box). Scale bar 5 µm. lg= lymph gland Genotypes are shown on top of corresponding panels. DAPI marks the nucleus.

DOI: https://doi.org/10.7554/eLife.18295.015

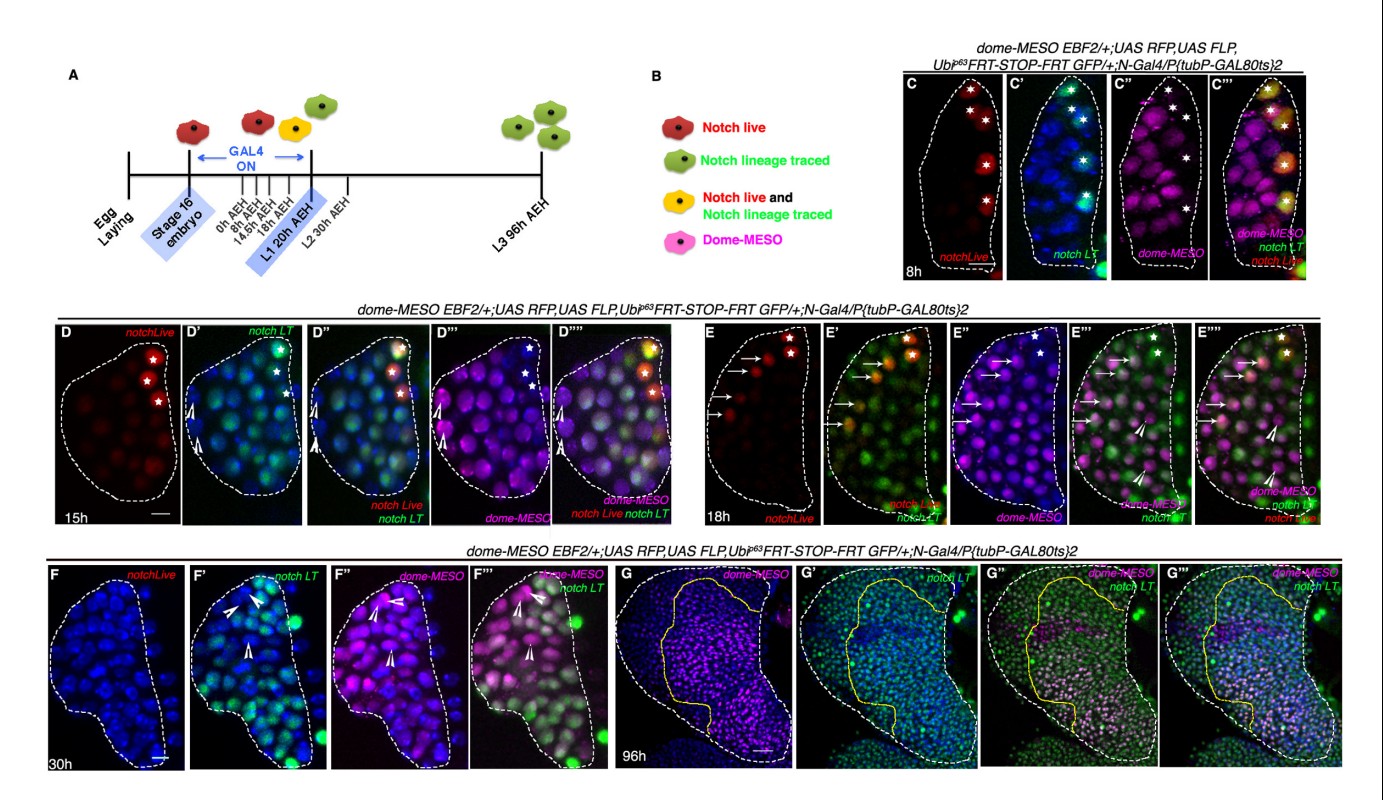

**Figure 5.** Hierarchical relationship between *dome* and *Notch* expressing cells in the first instar lymph gland. (**A**) Show the scheme of *N* lineage tracing followed throughout this panel. (**B**) Schematic representation of predicted cell types arising from *N* lineage tracing in **A**. A red cell indicates live expression of *N*. A green cell indicates that *N* was active in this or its precursor cell but is not active now. A yellow cell indicates continuous activation of *N>GTRACE* indicating that this cell expressed *N* earlier and is still expressing. A magenta cell indicates *dome-MESO* expression. (**C–C'''**) First instar lymph gland showing mutually exclusive yellow (both *N* live and lineage traced expression; stars) and magenta (*dome-MESO-EBF2*) cells at 8 hr (n = 6). (**D–D'''**) also shows few *N* expressing cells (yellow; stars in **D–D'''**) that do not express *dome-MESO-EBF2* (magenta; **D'–D'''**; n = 7) at 15 hr. (**E–E'''**) At 18 hr, some *N* expressing cells (arrows; **E–E'''**; n = 6) also express *dome* (arrows; **E''–E'''**), while some do not (stars; **E–E'''**), indicating a possible differentiation at this time point. (**F–F'''**) Second instar early (30 hr AEH) lymph gland does not actively express *N* (absence of red live expression in **F**) but contains *N* lineage traced cells (green; **F', F'''**; n = 5), which are now also expressing *dome-MESO* (magenta; **F''–F'''**). Also noteworthy are few cells (arrowheads) that are not lineage traced for N (not green) but still express *dome* (arrowheads, **D–F'''**). (**G–G'''**) *dome-MESO-EBF2* (magenta; **G, G''–G'''**) expression in restricted *N* lineage (green; **G'–G'''**) traced lymph gland following scheme in **A**. Scale bar is 20 μm for **G–G'''** for rest it is 5 μm. Genotypes are shown on top of corresponding panels. TOPRO marks the nucleus.

DOI: https://doi.org/10.7554/eLife.18295.016

The following figure supplement is available for figure 5:

**Figure supplement 1.** Lineage tracing in an identical window reveals the hierarchical relationship between *dome* and *Notch* expressing cells.

DOI: https://doi.org/10.7554/eLife.18295.017

*Figure 5A*. In order to trace the relationship between these two cell types, we assayed the larval lymph gland of the above genotype, not only in late third instar but also earlier at different developmental time windows. The stages were selected on basis of the proliferation profile discussed in *Figure 2* . In the above genotype, in addition to red (*Notch* real time expression) and green (*Notch*-lineage traced), magenta cells are seen that represent *dome-MESO* expression (*Figure 5B*).

At 8 hr AEH, we detected two kinds of cells: the first group consisted of 4–5 *Notch* expressing (stars, Red: *Figure 5C*) lineage traced (stars, Green: *Figure 5C'*) cells that were negative for *dome-less* (magenta) (*Figure 5C''–C'''*) and the other group expressed only *domeless* (magenta: *Figure 5C''*). Interestingly, the *dome* cells were not lineage traced for *Notch* (*Figure 5C''–C'''*), validating that indeed at this time point, none of the *Notch* expressing cells have undergone cell division. However, on analyzing 15 hr AEH three different cell types were observed. As before, the first type of cells expressed *Notch* (red and green thus yellow, stars) but lacked *domeless* (magenta)

expression. The number of these cells was now reduced to 3 (stars, *Figure 5D–D''''*). We have previously showed that cell division in *Notch* expressing cells starts from 13 hr onwards (*Figure 2I–I''*, N and *Figure 2—figure supplement 1G*). Interestingly, the second cell type observed was *Notch* lineage traced and *dome* positive (magenta) which also validated that indeed cell division has happened during this time. This group thus was the clonal expansion of *Notch* expressing cells that have now turned on *domeless* expression. In addition, we also detected few cells that were not *Notch* lineage traced but expressed *dome* (magenta: arrowheads). This group we believe, arose from the *dome* expressing cells that co-existed with *Notch* cells during the time of lineage tracing.

The numbers of *Notch* expressing cells were further reduced by 18 hr AEH (stars) with a concomitant increase of Notch lineage traced *dome* expressing cells (*Figure 5E–E''''*), clearly supporting our previous conclusion (arrows indicate cells that have low levels of Notch but have now up regulated *dome* expression). Remarkably, at 30 hr AEH the lymph gland was substantially populated by *dome* (magenta) expressing cells that were lineage traced for *Notch* (*Figure 5F–F'''*). Quite interestingly, none of the *Notch* (red: *Figure 5F*) expressing cells were seen from this time onwards, undoubtedly strengthening the fact that *Notch* positive cells eventually become *dome* positive cells. When this genotype was analyzed at late third instar (following the same regime of lineage tracing as in *Figure 5A*), this transient labeling of the *Notch* positive cells only during first instar resulted in labeling of almost all cells of the lymph gland including *dome* expressing progenitors (*Figure 5G–G'''*, also see *Figure 3C–C''*).

Put together, the above results clearly establish the hierarchical relationship of *dome* positive cells with respect to *Notch* positive cells.

We further endorsed the above finding by transient-lineage tracing of *dome* and *tepIV* expressing progenitors, following the same scheme that was employed for *Notch* (*Figure 5—figure supplement 1A* and *Figure 5A*). In case of *N-Gal4,* the transient activation of the lineage-tracing construct led to substantial labeling of the lymph gland (*Figure 5—figure supplement 1B–B''* and *Figure 3C–C'*) including the *dome* positive and Cubitus interruptus$^+$ (Ci$^+$) hemocyte progenitors. Compared to this, activation of the lineage-tracing cassette with *dome-Gal4* within the same short window resulted in generation of extremely restricted clonal expansion of *dome* expressing cells (*Figure 5—figure supplement 1D–D''*). This clearly showed that not enough multi-potent progenitors were born at that time to render labeling of the entire gland and that the potency of *Notch* expressing cells are higher than the *dome* positive ones. This was further validated by Ci expression (a specific marker for progenitors) that marked a large population of progenitors that were not lineage traced (Ci: red; *Figure 5—figure supplement 1D–D''*). Using *tepIV-Gal4,* another independent driver for hemocyte progenitors (*Figure 5—figure supplement 1E–E'*), a result akin to transient *domeless* activation was obtained (*Figure 5—figure supplement 1F–H*).

In contrast, the few multi-potent *Notch* expressing cells present at time had the potential to substantially label the almost entire lymph gland including *domeless* and Ci expressing progenitors.

In order to complement the lineage tracing experiment and provide a functional correlation, we induced genetic ablation of *Notch* expressing cells. This was achieved by the activation of *reaper* (*rpr*, pro-apoptotic gene) specifically in the HSCs by *Notch-Gal4*. As Notch is required in several development processes, we encountered early lethality on activation of *rpr*. A narrow window of four hours activation of *reaper* post emergence of larvae yielded few individuals that were not lethal at late third instar stage. To assay, if Notch expressing cells were successfully ablated, *dome-MESO-LacZ* was brought in the background of *N-Gal4*. On restricted activation of *UAS-rpr*, few individuals of every batch were analyzed at first instar stages to ensure that the lymph gland contain only *dome-MESO-LacZ* and no Notch expressing cells, confirming a successful ablation (Compare *Figure 6A* with *Figure 6C*). The siblings of the batch that had only *dome* expressing progenitors in the first instar lymph gland and made it to third instar stage, were analyzed. We found that genetic ablation of Notch expressing cells caused a massive reduction in the lymph gland size at third instar stage (Compare *Figure 6B* with *Figure 6D*). Quantitative analyses revealed a 5-fold reduction in the total size of the lymph gland (*Figure 6E*). Moreover, the ratio of area of progenitor cell population versus the total lymph gland area had dropped by more than 2 fold (*Figure 6F*) indicating that by killing the Notch expressing founder cells, the reserve population of progenitors arising from them were also eliminated. This in turn resulted in a massive reduction in the total lymph gland size that now housed the clonal expansion of those progenitors that coexisted with the HSCs.

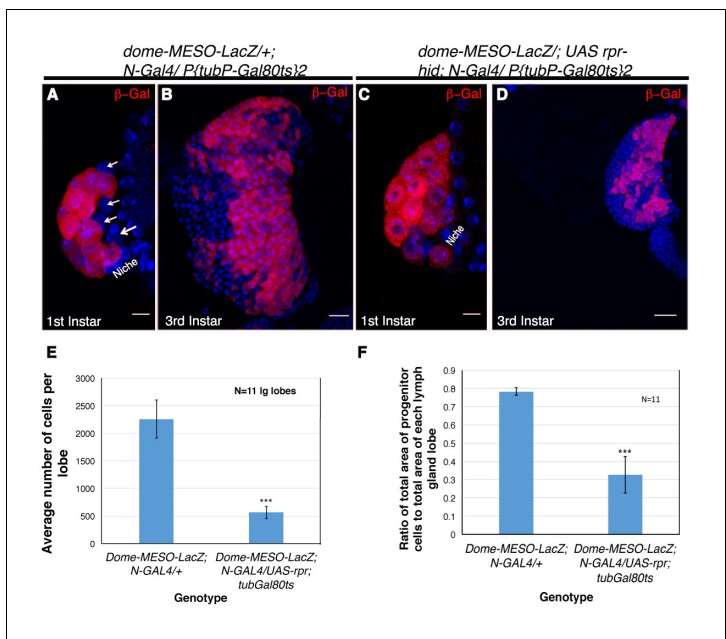

**Figure 6.** Genetic ablation of Notch expressing cells affects the size of the lymph gland. (**A**) Expression of *dome-MESO-LacZ* in the lymph gland 16 hr after hatching. Four cells (arrow) lacking *dome-MESO-LacZ* expression are seen. (**B**) *dome- MESO-LacZ* expression in third instar larval lymph gland. (**C**) Activation of Reaper (rpr) by *N- Gal4* in a short window of 4 hr resulted in elimination of the Notch positive HSCs. (**D**) Rearing the larvae of above genotype to third instar stages revealed a dramatic reduction in size of the lymph gland. (**E–F**) Quantitative analysis of average number of cells in the lymph glands of *UAS-rpr/ MESO-LacZ; tubGal80$^{ts}$; N-Gal4* revealed a 5 fold reduction in comparison to wild-type (**E**, 2.94016E-10; n = 11, two tailed unpaired Student's t-test). Similarly we also observed a 2.4 fold reduction in progenitor cell index (measured as the ratio of average area of Dome expressing cells with respect to total area of the lymph gland lobe) in comparison to the sibling control (**F**, p=1.2172E-08, two tailed unpaired Student's t-test, n = 11). Scale bar is 20 µm in **B** and **D** and 5 µm in **A** and **C**. Genotypes are shown on top of corresponding panels. DAPI marks the nucleus. Hours after larval hatching are as indicated in each panel. Error Bars=S.D.

DOI: https://doi.org/10.7554/eLife.18295.018

The following source data and figure supplement are available for figure 6:

**Source data 1.** Contains numerical data plotted in *Figure 6E,F*.

DOI: https://doi.org/10.7554/eLife.18295.020

**Figure supplement 1.** HSCs and progenitor lineages in the lymph gland.

DOI: https://doi.org/10.7554/eLife.18295.019

Put together, we can conclude that the *Notch* positive cells are higher in the order to *Dome* expressing progenitors and that the transient *Notch* expressing cells of first instar lymph gland are the multi-potent founder cells or the HSCs (*Figure 6—figure supplement 1*).

## Dpp signal from the PSC maintains the self-renewal of HSCs

We next attempted to determine the signal required for the maintenance of these transient HSCs. It has been demonstrated that the mature third instar larval lymph gland maintains its prohemocytes by Hedgehog (Hh) signaling from the adjoining PSC, which acts as its niche (*Mandal et al., 2007*; *Tokusumi et al., 2010*). However, we found that *hedgehog* expression is initiated in the niche only after 18 hr AEH (*Figure 7A*). This indicated that the known instructive role of Hh signaling from the niche in progenitor maintenance is a requirement that commences around late first instar stage. We next wanted to know whether the Antennapedia expressing hematopoietic niche of the early first instar larval lymph gland is mature and functional.

To probe this aspect, we checked for Lamin C (mammalian Lamin A/C homologue), a marker for mature or differentiated cells (*Riemer et al., 1995*). We found Lamin C expression in the first instar larval hematopoietic niche (marked by *antp-GFP*) even before *Hh* expression sets in (*Figure 7B–C'*).

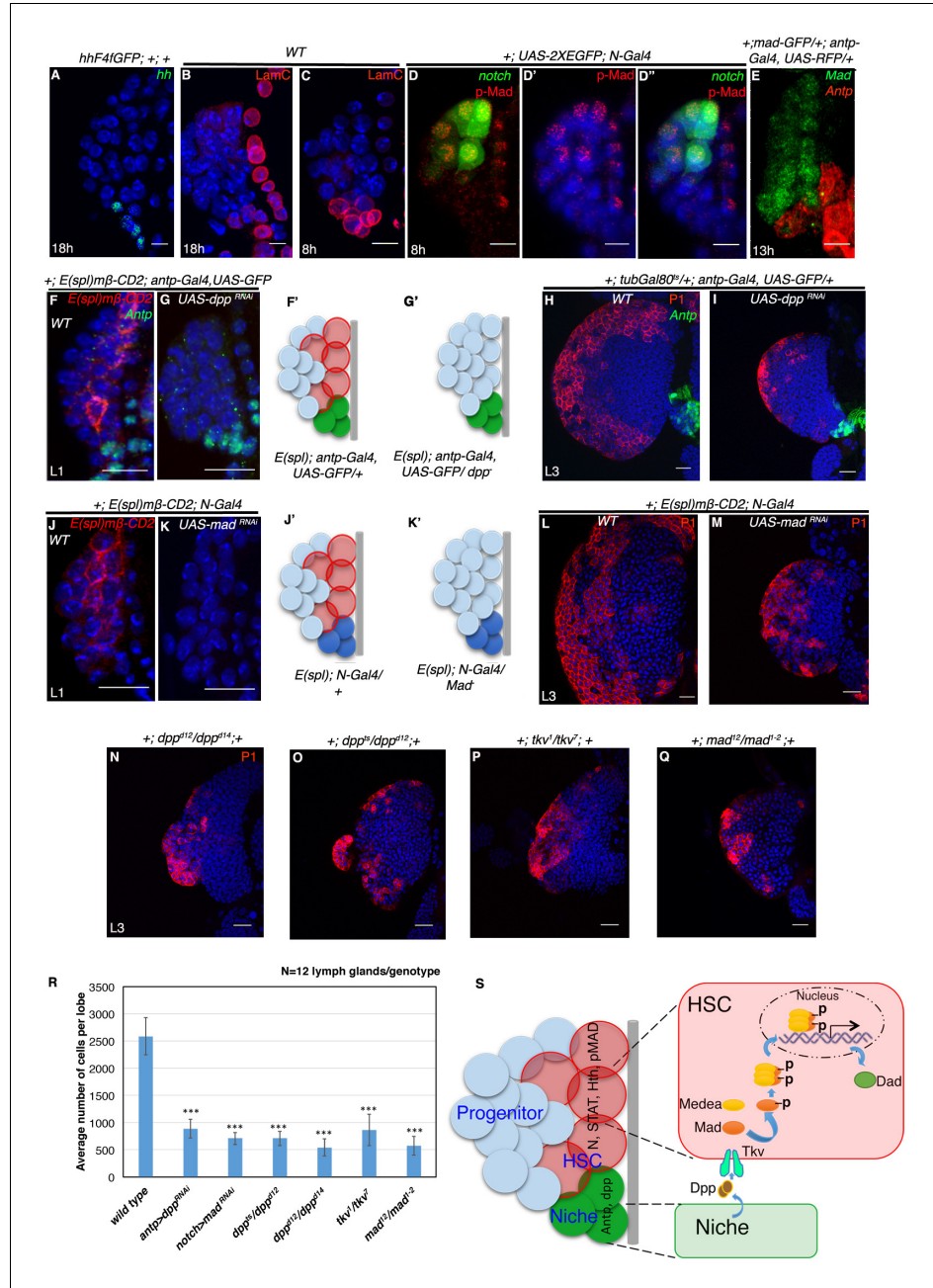

**Figure 7.** Requirement of Decapentapelagic (Dpp) from the PSC for hematopoietic stem cell maintenance in *Drosophila*. (**A**) The maintenance signal for progenitors: Hedgehog (green; n = 7) starts to express in lg at about 18 hr AEH. (**B–C'**) Lamin C (red) is expression in the niche/PSC (green) in 8 hr and 18 hr lg. (**D–D"**) pMad (red) labeling at 8 hr AEH is enriched in HSCs (*N-GFP*; n = 6). Also see *Figure 7—figure supplement 1C–C"*. (**E**) shows that *MadGFP* expression is predominantly absent from the PSC (**F–G'**) In first instar lymph gland, HSCs can be visualized by *E(spl)mβ -CD2* (red, [**F**]). Downregulation of Dpp function in the niche causes precocious loss of this HSC marker (E(spl), red, [**G**]). **F'–G'** represent scheme of results from (**F–G**). Also see *Figure 7—figure supplement 1D*. (**H–I**) shows that in a late third instar, above genotype causes a 3-fold reduction in the size of the lymph gland (P1, red, n = 15, p=3.98092E-10, two tailed unpaired Student's t-test) in comparison to the control in **H**. However, CZ cells (P1 positive: plasmatocytes, red, [**H–I**]) are present in above genotype. (**J–K'**) Attenuation of Mad expression by *Mad[RNAi]* in the HSCs (**K**) also causes premature loss of HSCs (*E(spl)mβ-CD2*, red, N = 15; 18 hr). **J'–K'** represent scheme of results from (**J–K**). Also see *Figure 7—figure supplement 1D*. (**L–M**) depicts that this genotype in a third instar stage also results in 3.6 fold reduction in the lymph gland size in comparison to the control in (**L**), although P1 positive cells are still detectable (red, n = 15, p=7.27E-10, two tailed unpaired Student's

*Figure 7 continued on next page*

*Figure 7 continued*

t-test; [**M**]). Also see **Figure 7—figure supplement 1E and G** and **Figure 7—figure supplement 1K–L**. (**N–Q**) The hetero-allelic mutant combination *dpp$^{d12}$/dpp$^{d14}$* (n = 12; p=4.75E-11, two tailed unpaired Student's t-test; [**N**]) or temperature sensitive mutant combination (*dpp$^{ts}$/dpp$^{d12}$*; n = 12; p=3.76329E-10, two tailed unpaired Student's t-test; [**O**]) causes a 3.6 and a 4.7 fold decrease in the size of the lymph gland respectively. Dpp receptor Thickveins (Tkv) mutant animals (*tkv$^1$/tkv$^7$*; n = 12; p=7.78E-11, two tailed unpaired Student's t-test; [**P**]) as well as Mad deficiency (*mad$^{12}$/mad$^{1-2}$*; n = 12; p=3.81811E-11, two tailed unpaired Student's t-test; [**Q**]) exhibit a similar decrease in the size. Like RNAi genotypes, in all classical loss of function, an analogous phenotype is seen. (P1, red; [**N–Q**]; compare with [**L**]). Also see **Figure 7—figure supplement 1H–J** (**R**) Quantification of the results from H–I and L–Q. Average numbers of cells per lobe are indicated. (**S**) Schematic representation of Dpp function in HSC maintenance. Dpp from the Antp expressing PSC is transported to pMad expressing HSCs (also expressing STAT, N, Hth), near the dorsal vessel to activate its receptor Tkv, leading to the nuclear translocation of Mad that maintains HSCs. Thus, loss of either *Dpp* from the PSC (**G**) or *Mad* (**K**) or loss of *Tkv* (**P**) from the *N* expressing cells results in precocious loss of HSCs. Scale bar = 5 µm for A-E and 20 µm for rest. Error bars=S.D. **Figure 7** has one figure supplement.

DOI: https://doi.org/10.7554/eLife.18295.021

The following source data and figure supplement are available for figure 7:

**Source data 1.** Contains numerical data plotted in **Figure 7R** and **Figure 7—figure supplement 1D,M**.
DOI: https://doi.org/10.7554/eLife.18295.023

**Figure supplement 1.** Involvement of Dpp signaling in HSC maintenance.
DOI: https://doi.org/10.7554/eLife.18295.022

---

This was a clear indication that these cells are not immature, but rather are in a differentiated state. We then speculated if this Lamin C positive early niche was in some way regulating the HSCs. This prompted us to look for unique paracrine signals that could maintain the newly identified *Notch* expressing cells at this stage.

We found *STAT-GFP* expression in both the HSCs and PSC from early first instar while its ligand reported by *upd3-GFP* initiated in only two PSC cells around 18 hr AEH (**Figure 1H** and **Figure 7— figure supplement 1A**). Wingless, another paracrine signal however, was present throughout the lymph gland from early first instar (**Figure 7—figure supplement 1B**). Hypothesizing that the spatial expression pattern of a maintenance factor should be limited to one population rather than covering both HSCs and PSC, we continued our search for a unique signal.

Dpp signaling, a key player in various stem cell scenario was examined next. Using an antibody against pMad (transcriptional activator of Dpp signaling), we could see enrichment of pMad expression in the Notch positive HSCs (**Figure 7D–D''** and **Figure 7—figure supplement 1C–C''** (**Affolter and Basler, 2007**; **Raftery et al., 1995**). Simultaneous detection of Antp (*antp-Gal4, UAS-mCD8RFP*) and pMad (*Mad-GFP*) demonstrated insignificant levels of Mad in the first instar PSC (**Figure 7E**), further implicating the possibility of PSC being the source of Dpp. To address this ,we generated a fly line that harbours the notch reporter *E(spl)mβ-CD2* in the background of *antp-Gal4, UAS-GFP* and used this to downregulate Dpp signaling from the PSC by expressing *UAS-dppRNAi* and study its effect if any, on HSCs by visualizing the expression of *E(spl)mβ-CD2*. As evident from **Figure 7F,F' and G,G'**, expressing *UAS-dpp RNAi* completely abolished *E(spl)mβ-CD2* expression, suggesting a premature loss of Notch positive HSCs.

We next assayed the long term effect of the loss of these founder cells on the mature lymph gland. Using *Gal80$^{ts}$* to restrict the *Gal4* activity and thereby the loss of Dpp only in a narrow window (stage 16 of embryogenesis to 18 hr AEH), we observed a 3 fold decrease in the size of the lymph gland when compared to the controls (Compare **Figure 7H** with **Figure 7I**). We reasoned that if Dpp is indeed the signal from the niche, attenuation of downstream transcription factor Mad (Mothers against Dpp) in the receiving cell should elicit an analogous response. Consistent with our rationale, we found that downregulation of Mad expression specifically from the *Notch* positive cells, also abolished the *E(spl)mβ-CD2* expression in first instar larval lymph gland (Compare **Figure 7J, J'** with **Figure 7K,K'**). This eventually resulted in significant reduction in the size of late third instar larval lymph gland (Compare **Figure 7L** with **Figure 7M**). We further analyzed the distinct nuclear size (**Figure 1R**) of the first row of cells near the dorsal vessel of the above genotypes at first instar stage. Quantitative analyses of the nuclear size exhibited a significant reduction when compared to the control (**Figure 7—figure supplement 1D**), validating the loss of *Notch* positive HSCs.

Thus, loss of Dpp from the PSC or Mad from the HSC during first instar affects the sustenance of the founder cells. This in turn majorly reduces the progenitor reserve, which affects the overall size of the lymph gland at later stages, a phenotype analogous to genetic ablation of HSCs (*Figure 6*). This is also in tune with our previous observation that demonstrated that the Notch negative cells residing in the first instar larval lymph gland, also end up in populating the mature lymph gland (*Figure 3C–E''and Figure 6—figure supplement 1).*

We further confirmed the RNAi knockdown results using various mutant allelic combinations of *dpp*, *tkv* (*thickveins*, Dpp receptor) and *Mad* (*Figure 7N–Q*). Quantitative analyses of the average number of cells in the lymph gland of hetero-allelic combination $dpp^{d12}/dpp^{d14}$, temperature sensitive $dpp^{hr56}/dpp^{d12}$, $tkv^1/tkv^7$ and $Mad^{12}/Mad^{1-2}$ revealed a 4.8, 3.6, 3 and 4.5 fold reduction when compared to wild type lymph gland, respectively (*Figure 7R*).

All these results clearly demonstrate that the first instar PSC is a functional entity that releases Dpp required for HSC maintenance. Interestingly, in all the above mutant genotypes, although the relative size of the lymph gland was small, it had both cortical zone (P1:Nimrod marks the differentiated cells) and medullary zone (Ci: Cubitus interuptus marks the progenitor cells) (*Figure 7H,I,L,M–Q* and *Figure 7—figure supplement 1E–J*). It was compelling to observe that the expression of E (spl)*mβ-CD2* in the CZ (co-localizes with *Hml-GFP*, a differentiated hemocyte marker; *Figure 7—figure supplement 1K–K''*) was not affected in Mad loss of function condition (*Figure 7—figure supplement 1L*).

Previously, it has been shown that Dpp signaling is directly involved in cardiogenic mesoderm formation and thereby indirectly in lymph gland generation (*Mandal et al., 2004*). Another study revealed that Dpp signaling is required in the third instar larvae to regulate the size of PSC (*Pennetier et al., 2012*). Here in this current study, we effectively demonstrate that Dpp signaling is also a requisite during the first instar larvae to sustain and maintain the HSCs existing at that time without affecting the niche. Quantitative analysis revealed that although the niche number got affected in *dpp* mutants, the loss of Dpp in the first 18 hr AEH, however, affected the HSCs only (*Figure 7—figure supplement 1M*) as the niche cell numbers in this case was comparable to wild type. Thus  ctivation of this pathway leads to nuclear translocation of Mad (*Figure 7S*) in the *E(spl) mβ-CD2* HSCs, which in turn maintains them. Dpp signaling has never been implicated in non-autonomous control over any other cell type in the lymph gland directly. Our study highlights this aspect of Dpp signaling in the homeostasis of the larval lymph gland. In this context, it is worth noting that both male and female germline stem cells in *Drosophila* also employ Dpp pathway for their maintenance (*Xie and Spradling, 2000*; *Kawase et al., 2004*).

## Discussion

Our substantial understanding of definitive hematopoiesis in *Drosophila* is largely restricted to embryonic and third instar larval stages of the lymph gland. Here, in this study, we have ventured into the early first instar larval lymph gland and discovered the founders of the considerably well studied prohemocytes that populate the MZ. These HSCs express several molecular markers, which are commonly associated with hematopoietic stem cells in vertebrates namely Notch, STAT, Homothorax and BMP transcription factor Mad (*Koch et al., 2013*; *Stine and Matunis, 2013*; *Argiropoulos et al., 2007*; *Unnisa et al., 2012*; *Varnum-Finney et al., 1998*; *Burns et al., 2005*). Anatomical analysis of the HSCs unravels its close association with dorsal vessel (DV), an equivalent of larval heart. This is in tune with previous observation in zebra fish and mouse, where it has been established that HSCs reside primarily in the vascular and/or perivascular niches during embryonic and fetal development (*Kiel and Morrison, 2008*). Together, these results highlight the conserved anatomical location preferred for HSC formation across divergent taxa.

Like any other stem cell population (*Hsu and Fuchs, 2012*), the HSCs in the first instar larval lymph gland are also niche dependent. They rely on Dpp signaling from the previously identified hematopoietic niche/PSC. Our genetic analyses clearly demonstrate that attenuation of Dpp signaling affects HSCs maintenance in the early lymph gland. This reflects a great degree of similarity to the vertebrate AGM related HSCs that express BMP4 receptor and require BMP4 signaling as the niche signal from the neighboring mesenchyme for their maintenance (*Durand et al., 2007*; *Drevon and Jaffredo, 2014*). It has been demonstrated that both the vertebrate AGM and *Drosophila* cardiogenic mesoderm are specified by identical signals and house blood cell precursors

originating from hemangioblasts (*Mandal et al., 2004*; *Choi et al., 1998*). It is interesting to note that the site of HSC genesis: the lymph gland, itself arises from cardiogenic mesoderm, a region analogous to vertebrate AGM.

In vertebrates, Hematopoietic Stem Cells (HSCs) undertake a long journey that involves multiple sites like yolk sac, the AGM region, the placenta and the fetal liver for their maturation and expansion finally colonizing into bone marrow (*Mikkola and Orkin, 2006*). Strikingly, the blood precursors present in the lymph gland upon expansion actually home into the adult hematopoietic hub, a simpler version of the bone marrow (*Ghosh et al., 2015*). However, in contrast to vertebrate system, in case of *Drosophila*, the larval lymph gland serves as the only organ involved for maturation and expansion of hemocytes before the final seeding happens into the hematopoietic hub of adult. We believe that since the entire process of genesis, expansion and lineage commitment happens within the lymph gland, the HSCs, unlike mammals, in this case are transient. Our temporal analysis reveal that that the *Notch* expressing HSCs can be observed in the first instar lymph gland only for the first 20 hr of development. During this time frame they divide and give birth to *domeless* expressing blood cell progenitors. Thus, it can be considered that the requirement of HSCs in the lymph gland is to establish a stockpile of undifferentiated progenitors that in turn generates differentiated blood cells for the immediate requirement of larval development and subsequently to support the upcoming events of metamorphosis and adult hematopoiesis.

With the identification of the HSCs in the early larval lymph gland, the current study not only extends the range of conservation shared between the two divergent taxa but also provides a new model of *Drosophila* lymph gland development that is more similar to earlier events of HSC genesis, maintenance and migration in mammalian hematopoiesis (*Figure 8A–B*).

Since there exist s resounding similarity between the early larval HSCs and vertebrate AGM related HSCs, we speculate that this newly identified model can be employed to gain a better insight into some developmental aspects associated with AGM derived HSCs. Extensive studies employing mammalian HSCs have addressed several aspects of HSC biology that include but are not limited to their developmental origin, molecular mechanisms regulating their cycling, viability and lineage commitment of HSCs as well as several issues related to hematopoietic abnormalities

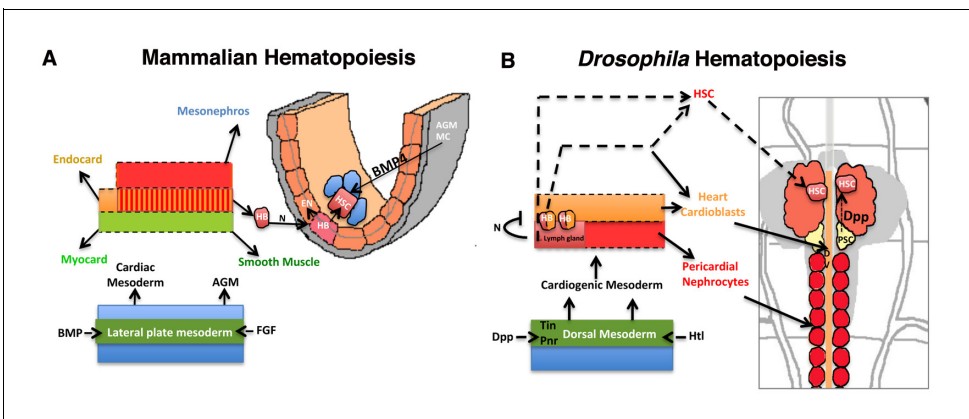

**Figure 8.** Significant parallels in HSC development in Mammalian and *Drosophila* hematopoiesis. Models depicting similarities in the development of HSCs from the mammalian AGM (**A**) and *D. melanogaster*'s cardiogenic mesoderm (**B**; redrawn from (*Mandal et al., 2004*). The cardiogenic mesoderm in *Drosophila* is analogous to the developing AGM of vertebrate and is nourished by identical signals. In vertebrates, a subset of AGM-derived cells has been proposed to constitute hemangioblasts (HB), which produce HSCs and endothelial cells. These HSCs also require BMP4 signaling from the AGM mesenchyme (AGM MC) for its maintenance. Likewise in flies, a seeding HB has been proposed to give rise to a subpopulation of cardioblasts and lymph gland primordia. Lymph gland progenitors are thus derived from few HSCs that also require niche-derived Dpp signaling for their maintenance. AGM: Aorta-Gonad-Mesonephros; EN: Endothelium; HSC: hematopoietic stem cell; FGF: Fibroblast Growth Factor; Heartless: FGF fly homolog Htl; AGM MC: AGM mesenchyme; FGF: fibroblast growth factor; BMP: Bone morphogen protein; Dpp: Decapentapelagic (BMP fly homolog); HB: Hemangioblast; PSC: Posterior Signaling Centre (Niche); DV: Dorsal vessel, heart.

DOI: https://doi.org/10.7554/eLife.18295.024

(*Ciau-Uitz et al., 2014*; *Dzierzak and Speck, 2008*; *Babovic and Eaves, 2014*; *Eaves, 2015*; *Golub and Cumano, 2013*; *Kim et al., 2013*; *Méndez-Ferrer et al., 2015*; *Mikkola and Orkin, 2006*; *Nakano et al., 2013*; *Orkin and Zon, 2008*; *Tavian et al., 2010*). One of the aspects that has drawn much attention is to understand the mechanisms regulating self-renewal of early HSCs as it holds promise in developing measures for early HSCs expansion. Due to technical limitations and rarity of AGM HSCs (*Medvinsky et al., 2011*), we still have little understanding about the nature of division that the cells undergo. Employing this amenable invertebrate model, we can now gather insights into such a process and boost our understanding of mechanism of early HSC expansion.

Several enthralling cell biological queries have emerged out of this current study. For instance, presence of Trio, a unique rho/rac1 dependent GEF in both HSCs and in the cells of PSC/niche, paves the foundation to explore the role of this unique GEF in AGM HSC development. Previous studies have highlighted the role of the mammalian Guanidine Exchange Factor (GEF) in yolk sac and fetal liver hematopoiesis as well as in homing of HSCs into the bone marrow (*Ghiaur et al., 2008*; *Satyanarayana et al., 2010*). But what role it might execute in AGM related HSCs has not been addressed.

Another important outcome of this study is the observation of the early expression of Notch in the HSCs that lie adjacent to the anterior aorta. Employing advanced genetic tools available in flies, it might be possible in future to genetically dissect the role of Notch in combination with other signaling molecules in the process that distinguishes between arterial and hematopoietic fate during development. Furthermore, identification of Dpp as the signaling molecule from the niche that is critical for HSC maintenance, in conjunction with the previously known role of the Hh signaling in maintenance of the progenitor cells of the medullary zone (*Mandal et al., 2007*), indicates that lymph gland can be employed as a great model to understand temporal role of these two different morphogens in regulating early hematopoiesis.

The identification of HSCs in *Drosophila,* thus, brings about a paradigm shift in our understanding of *Drosophila* hematopoiesis. Given the significant conservation in the early HSC development with vertebrates, the outcome of this study holds the promise of opening new avenues to better understand developmental hematopoiesis. This has a far-reaching implication spilling into stem cell based therapies, wherein it is imperative to know how tissue specific stem cells are specified in development. Furthermore, the transcriptional networks controlling HSC fate during mammalian embryonic development are highly complex (*Robin and Durand, 2010*). In this direction, the fly model can step in to identify and elucidate not only normal development but also several uncharacterized early pathological events linked with embryonic HSC related disorders.

## Materials and methods

### Genetics

All embryo collections were done for 4 hr on mixed fruit juice-agar-sugar medium with a little fresh yeast paste onto them. 1 hr larval batches were synchronized upon hatching and transferred on standard cornmeal/yeast/sugar medium to obtain a synchronized larval population. Such lymph glands were then dissected at appropriate developmental time points and processed for immunostaining. Detailed genotype of fly lines used in this study and their sources are described in Supplementary file 1.

### Lineage tracing

For lineage tracing experiments *Gal4* technique for real-time and clonal expression (G-TRACE) was used. GTRACE combines *Gal4-UAS*, FLP recombinase–FRT and fluorescent reporters to generate cell clones that provide information about the origins of individual cells in *Drosophila. GTRACE* in a *Gal4* dependent manner not only marks the current or the real time expression of the gene driving the *Gal4* with a RFP but also independently marks all the cells arising from the same gene lineage with a GFP. To initiate lineage tracing, FLP recombinase upon being activated in a *Gal4* dependent manner, flips out FRT-flanked transcriptional termination cassette between *Ubi-p63E* promoter and EGFP construct. This brings EGFP directly under the *Ubi-p63E* promoter thereby perpetuating all the progeny cells arising thereafter with EGFP.

A *UAS-G-TRACE/cyo* stock was crossed to *N-Gal4* both in the absence or presence of a *tub-Gal80ts* temperature sensitive transgene (McGuire et al., 2003) to allow temporal activation of the *Gal4*. For HSC lineage experiments, crosses were kept at 29℃ only from embryonic stage 16 to 18 hr AEH. Before and beyond this crucial timeline they were kept at 18℃ to activate the *Gal80* repressor. To check the lineage of only one or two *Notch* expressing cells, larvae were shifted to restrictive temperature (29℃) between 15 hr to 22 hr AEH. For estimating hierarchical relationship between *dome* and *Notch*, *dome-MESO-EBF2* was brought in combination with *N-Gal4* and shifted following the same regime as earlier. For a simultaneous negative control experiment to show *N* requirement in crystal cells, *N-Gal4>GTRACE; Gal80ts* cross was kept at 18℃ until the L2 stage and then transferred to 29℃ to deactivate the gal80 repressor. *UAS-GTRACE/Cyo; Gal80ts* was also crossed to *dome-Gal4* and *tepIV-Gal4* to analyze the lineage of hematopoietic progenitors in *Drosophila* lymph gland. For restricted *dome* and *tepIV* labeling, larvae were shifted to 29℃ only from stage 16 of embryogenesis to 20 hr AEH and then shifted back to 18℃ till mid late instars.

## Heat shock pulse experiments

For label retention assay, *UAS-Histone2B::YFP* (from F. Shweisguth (Bellaïche et al., 2001)) was crossed to a ubiquitous transient driver *hs-Gal4*. To label all cells of lymph gland, stg 16 embryos were heat shocked for 1 hr at 37℃ (pulse). To check if all cells were labeled at time 0, embryos were kept at 25℃ for 2 hr and fixed thereafter. To check retention of label (chase), embryos were kept at 25℃, allowed to hatch and further synchronized. Such larvae were then dissected at 12, 13 and 18 hr to check for loss of label.

## *Fucci* cell cycle analysis

*UAS-S/G2/M-Green fucci* (Sakaue-Sawano et al., 2008; Makhijani et al., 2011) fly line contains a fluorescent probe- mAG (monomeric Azami Green) fused to the deletion mutant of human Geminin. It shows fluorescence during the S/G2/M phases of the cell cycle and no fluorescence in G1 phase. Also, accumulation of S/G2/M-Green in the nucleus indicates that the cell is in the S/G2 phase while distribution of S/G2/M-Green into the cytoplasm corresponds to the initiation of M phase. *UAS-S/G2/M-Green fucci* fly stock was crossed to *N-Gal4* and *E(spl)mβCD2; N-Gal4* to ascertain the cell cycle status of HSCs.

*UAS-GFP-E2f1$_{1-230}$ UAS-mRFP1NLS-CycB$_{1-266}$* (Zielke et al., 2014) fly line depends on GFP and RFP tagged degrons from E2F1 and Cyclin B proteins, that are degraded by APC/C and CRL4$^{cdt2}$ ubiquitin E3 ligases, as they enter S phase and mid mitosis, respectively. It shows green fluorescence in G1 phase (due to accumulation of *GFP-E2f1$_{1-230}$*), red fluorescence in S phase (due to accumulation of *mRFP1NLS-CycB$_{1-266}$*), and yellow fluorescence in G2 and mitosis (due to presence of both *GFP-E2f1$_{1-230}$* and *mRFP1NLS-CycB$_{1-266}$*). *UAS-GFP-E2f1$_{1-230}$ UAS-mRFP1NLS-CycB$_{1-266}$* fly stock was crossed to *N-Gal4* to ascertain the cell cycle status of HSCs. Average number of cells at each stage was counted and plotted. All flies were kept at 25℃. Larvae were dissected at 8, 13, 15 and 18 hr AEH.

## Cell death assay

*UAS-apoliner5* contains two fluorophores- mRFP and eGFP linked with a caspase sensitive site. Upon caspase activation, the sensor is cleaved and eGFP translocates to the nucleus, leaving mRFP at membrane (Bardet et al., 2008). *UAS-apoliner5* was crossed to *N-Gal4* to detect if *N*-expressing cells are undergoing apoptosis. All flies were kept at 25℃. Larvae were dissected at 8, 13 and 18 hr AEH.

## Immunostaining and microscopic analysis

Lymph glands of 1 hr synchronized batches were dissected on ice in 1X PBS. The pull outs/tissues were fixed in 4% formaldehyde prepared in 1X PBS (pH 7.2) for 40 min followed by two quick washings of 1X PBS. Tissues were permeablized by 0.3% PBT (0.3% triton-X in 1X PBS) for 30 min (3 washings, 10 min each). Blocking was done in 10% NGS (prepared in 0.3% PBT) for 45 min-1hr. Tissues were then incubated in primary antibody with appropriate dilution in 10% NGS (prepared in 0.3% PBT) for 18–24 hr at 4℃. Primary antibody was washed by a quick wash of 0.3% PBT followed by 4 washes of 15 min each with 0.1% PBT for 1 hr. Tissues were again blocked with 10% NGS (prepared

in 0.1% PBT) for 20–30 min. Secondary antibody specific to primary antibody was added and kept at 4°C for 18–24 hr followed by washings with 0.1% PBT (15 min X 2) and 0.3% PBT (15 min X 2) and incubating them in DAPI solution for 1 hr at room temperature. DAPI was subsequently washed with 1X PBS and mounted in Vectashield (Vector Laboratories).

Primary antibodies used were mouse anti-Lamin C (DSHB Cat# lc28.26 RRID:AB_528339, 1:20), mouse anti-Antp, (DSHB Cat# anti-Antp 4C3 RRID:AB_528082, 1:5), mouse anti- Wg (DSHB Cat# 4d4 RRID:AB_528512, 1:3), rat anti-Ci (DSHB Cat# 2A1 RRID:AB_2109711, 1:3), mouse anti-Hnt (DSHB Cat# 1g9 RRID:AB_528278,1:5), mouse anti-trio (DSHB Cat#9.4Aanti-Trio RRID:AB_528494, 1:20), rabbit α-hth (*Noro et al., 2006*) (*Kurant et al., 1998*) 1:400; kind gift from R. Mann), rabbit anti-odd (1:500; kind gift from J. Skeath), rabbit anti-p-MAD, (PS2, (*Gancz et al., 2011*) (*Tanimoto et al., 2000*), 1:500; kind gifts from E. Laufer and T. Tabata), mouse anti-CD2 (Fitzgerald Industries International Cat# 10R-CD2cHUp RRID:AB_1283138, 1:200), rabbit anti-Serpent (α-Srp, (*Hayes et al., 2001*) 1:500; kind gift from D. Hoshizaki), mouse anti-β-Galactosidase (β-Gal, Promega Cat# Z3781 RRID:AB_430877, 1:100), mouse Anti-GFP (Sigma-Aldrich Cat# G6539 RRID:AB_259941, 1:50) Mouse anti-P1 (*Kurucz et al., 2007*; Cat# NimC1 RRID:AB_2568423), 1:50; kind gift from I. Ando) and Rabbit anti-Phospho Histone (Cell Signaling Technology Cat# 3642S RRID:AB_10694226 , 1:200).

The secondary antibodies used were: Alexa 488 goat anti rabbit (Thermo Fisher Scientific Cat# A11008 RRID:AB_143165) or FITC (goat anti mouse-Jackson ImmunoResearch Labs Cat# 115–095-166 RRID:AB_2338601 and goat anti rabbit-Jackson ImmunoResearch Labs Cat# 111–095-144 RRID: AB_2337978). Alexa 568 (goat antimouse-Thermo Fisher Scientific Cat# A-11004 RRID:AB_2534072, goat anti rabbit-Thermo Fisher Scientific Cat# A11011 RRID:AB_2534078 and goat anti rat-Thermo Fisher Scientific Cat# A11077 RRID:AB_10562719) were used at 1:500. Cy3 (donkey anti mouse-Jackson ImmunoResearch Labs Cat# 115–165-166 RRID:AB_2338692 and donkey anti rabbit-Jackson ImmunoResearch Labs Cat# 711–165-152 RRID:AB_2307443) were used at 1: 500. Dylite (donkey anti rat-Jackson ImmunoResearch Labs Cat# 712–605-150 RRID:AB_2340693 and goat anti mouse-Jackson ImmunoResearch Labs Cat# 115–605-003 RRID:AB_2338902) and Alexa 647 (goat anti mouse-Thermo Fisher Scientific Cat# A-21236 RRID:AB_2535805 and goat anti rat-Thermo Fisher Scientific Cat# A-21247 RRID:AB_141778) used at 1:500. DNA was stained with 4,6-diamidino-2-phenylindole (DAPI; Sigma) at 1 µg/ml and TOPRO at 1:500 in 1xPBS. For co-stainings with two antibodies, antibody with a stronger binding affinity was first developed followed by the second.

All microscopy were performed on a Zeiss LSM 780 confocal microscope. Images were acquired as confocal sections using the same settings within each set of experiments. Image J(RRID:SCR_003070) and Adobe Photoshop CS3 (RRID:SCR_002078) were used to assemble the images. Three-dimensional rendering was performed by Imaris software (RRID: SCR_007370).

All microscopy were performed on a Zeiss LSM 780 confocal microscope. Images were acquired as confocal sections using the same settings within each set of experiments. Image J and Adobe Photoshop CS3 were used to assemble the images. Three-dimensional rendering was performed by Imaris software.

## Image analysis

Every experiment was repeated at least thrice to check for reproducibility. Usually, more than ten randomly selected lymph glands were analyzed per genotype, and the statistical significance was calculated with Two tailed unpaired Student's t-test.

12 bit confocal sections (1 µm) of *N-Gal4, UAS-2XEGFP* with unsaturated signal were used to generate additive projections with ImageJ. Total GFP fluorescence intensity per cell was measured as mean fluorescence intensity × total cellular area. High GFP intensities of 45 HSCs in three different experiments were compared to low GFP intensities of first batch of progenitors and plotted as fold change. Based on this, count and total area of HSCs and first batch of progenitors were measured at different developmental time points and plotted. For each of these analyses, at least 3 biological replicates and 4 technical replicates of the experiment was conducted.

## Acknowledgements

We thank I Ando, U Banerjee, E Bach, S Cohen, J Skeath, D Hoshizaki, E Laufer, R Mann, M Miura, K Moberg, S Noselli, A Salzberg, F Shweisguth, A Tsakonas and T Tabata for reagents. We thank all

members of the two labs for their valuable inputs. We thank IISER Mohali's Confocal Facility, Bloomington *Drosophila* Stock Center, Indiana University and Developmental Studies Hybridoma Bank, University of Iowa for flies and antibodies. Wellcome DBT Intermediate Fellowship to LM and Institutional support to SM and NSD funded this study.

## Additional information

### Funding

| Funder | Grant reference number | Author |
| --- | --- | --- |
| Wellcome Trust DBT Alliance | 500124/Z09/Z | Lolitika Mandal |
| Indian Institute of Science Education and Research Mohali | | Nidhi Sharma Dey<br>Parvathy Ramesh<br>Mayank Chugh<br>Sudip Mandal<br>Lolitika Mandal |

The funders had no role in study design, data collection and interpretation, or the decision to submit the work for publication.

### Author contributions

Nidhi Sharma Dey, Acquisition of data, Analysis and interpretation of data, Drafting or revising the article; Parvathy Ramesh, Acquisition of data, Drafting or revising the article; Mayank Chugh, Involved in initial characterization and standardization of histone labeling, Acquisition of data; Sudip Mandal, Analysis and interpretation of data, Drafting or revising the article; Lolitika Mandal, Conception and design, Analysis and interpretation of data, Drafting or revising the article

### Author ORCIDs

Lolitika Mandal ⓘD http://orcid.org/0000-0002-7711-6090

### Decision letter and Author response

Decision letter https://doi.org/10.7554/eLife.18295.026
Author response https://doi.org/10.7554/eLife.18295.027

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
