## [Decision Letter]

Thank you for submitting your article "Dpp dependent Hematopoietic Stem Cells give rise to Hh dependent blood progenitors in larval lymph gland of *Drosophila*" for consideration by *eLife*. Your article has been favorably evaluated by K VijayRaghavan (Senior Editor) and three reviewers, one of whom, Yukiko M Yamashita (Reviewer #1), is a member of our Board of Reviewing Editors.

The reviewers have discussed the reviews with one another and the Reviewing Editor has drafted this decision to help you prepare a revised submission.

Summary of review:

In this study, Dey at al., describe the presence of previously unreported multi-potent stem cells of hemocytes in the *Drosophila* first instar larval lymph gland. Although past studies have shown the presence of progenitors, the presence of "hematopoietic stem cells" (HSCs) in *Drosophila* has not yet been fully established. The authors use cell specific markers and morphometric measurements to determine the presence of a unique cell type in proximity to the dorsal vessel of the first instar larvae. These cells express high levels of serpent (srp), a pan-blood cell marker, but lack a prohemocytes marker, domeless (dom). Using a candidate approach they discover that these cells are found 8 hours after egg hatching (AEH), are larger than prohemocytes and express high levels of Notch. They are initially slow cycling and eventually enter mitosis 13 hours AEH. They complete division by 18 hours AEH to give rise to progenitors of the larval lymph gland, which eventually express Dome. The authors further establish that the posterior signaling center (PSC) of the lymph gland acts as the HSC niche and provide Dpp signaling for the maintenance of these cells. Loss of Dpp signaling, results in the failure of HSC self-renewal and in the formation of small lymph glands. Altogether, these results indicate the presence of HSCs in the first 20 hours of larval development that establishes a reservoir of hemocyte progenitors required for adult hematopoiesis.

This is an important study as the authors characterize a novel stem cell population and identify a signaling pathway in the niche that maintains these cells. They carry out beautiful lineage tracing experiments that show the existence of HSCs and the cells they give rise to.

Essential revisions:

Several points were raised that have to be addressed prior to publication. The reviewers agreed that they only involve relatively straightforward revision experiment(s) and text editing. Thus, we would like to invite you to submit a revised version. Requested revisions are outlined below.

1) Lineage tracing experiments: in order to show the multipotency, the clones should be induced at low frequency and ideally at single cell level, and then show that a single labeled HSC can expand and differentiate into different cells in this lineage. In complementary, these potential HSCs could be genetically ablated and show no progenies arising from them. Along the same line, in Figure 5, the authors made a strong statement that those N-expressing cells are founder cells for those Dome-expressing cells. However, some of the later stage Dome-expressing cells also come from N-negative cells. It is OK that more than one founder cell exists. But in order to complement the tracing experiments, more convincing evidence would be ablation of those N-positive cells lead to significant decrease of Dome-cells, similar to the experiments in Figure 6. If these experiments are proven to be technically challenging, please elaborate on this point in the response letter.

2) The overall characterization of co-expression vs. mutually exclusive patterns in Figure 1 should be done with co-labeling. If the figure only has one staining, it is hard to make such a claim, such as in Figure 1C, 1E-G, H, etc.

3) They state that HSC expresses STAT: this is somewhat puzzling given that the progenitor cells are Dome-positive: the upstream receptor for STAT. What is the relationship between Dome+ cells and STAT+ cells? (if they are distinct cell types as inferred by authors, then Dome+ cells are not transducing STAT signaling? And how STAT+ cells can become STAT+ without expressing Dome?)

4) The authors report that the Notch expressing cells in the earlier time points are bigger in size, than the neighboring cells. And these cells diminish over time. However, the data presented to support this is not convincing. The Notch expressing cells in Figure 1M look much bigger than the ones in Figure 1N (1M scale bar is missing). Do these cells diminish in size over time? Are they bigger in 8h time point when compared to 13h. A quantitative analysis of the Notch expressing cell size over time should address this concern.

5) Dome-meso-lacZ positive cells in Figure 1B" also have different level of lacZ, any explanation for that? Also, it seems that Dome-meso-lacZ and dome-Gal4: UAS-GFP do not have totally overlapping signal in Figure 1—figure supplement 1C-C", why?

Additional points:

Other comments have been raised regarding the clarity of the manuscript. Although these are not about the scientific contents of the manuscript, we recommend that these points are addressed prior to publication.

1) The manuscript can be improved by editing. Please try to increase the clarity of the manuscript to appeal to a broader readership of *eLife*.

2) The figures and figure description need work, as there are some panels that are hard to decipher. Examples are:

The figure panels are not necessarily in temporal order e.g. Figure 6 B-C.

Also the Figure 4A-B' lettering is not aligned. Figure 1Q is not necessary and can be mentioned in the text?

Figure 1B and Figure 1—figure supplement 1A-A' show the same data and should not be repeated.

The scale bars between Figure 2—figure supplement 2A and Figure 2—figure supplement 2B are not the same. But they seem to be of the same magnification?

In Figure 6C a co-stain of LaminC with Dot/Antp would better ascertain their identity as PSC niche cells.

Figure 6D-D" and Figure 6—figure supplement 1C-C" are repetition of the same data, with the latter not being mentioned in the text.

3) A little more information on Odd expression will help a broad audience understand this better.

4) Does DPP affect size of the niche itself? They should monitor the niche size to see if the effects are direct or indirect. Also, how will Dpp/TKV overexpression effect the HSC population?

5) Notch staining should be included in Figure 6E to show the range of niche-to-stem cell signaling in this system. Are the "bigger" high-level Notch expressing cells closer to the PSC?

6) The dual FUCCI system is easier to interpret in distinguishing between the cell cycle phases than the Geminin FUCCI reporter. The supplemental data that shows the dual reporter could be moved to the main figure. Also in Figure 2—figure supplement 1F, the authors distinguish between G2 and M phase cells using Dual FUCCI system. This seems inappropriate, as both phases would have a "yellow" nuclear stain. They can address this by staining for PH3 in the dual FUCCI system. The graph distinguishes these two cell cycle phases but the reporter does not. Did they do a PH3 in this system as well? It is not shown.

7) Figure 2: the term "label retaining cell" is confusing. What the data show in this figure is actually histone label is simply diluted as N+ cells undergo division. Of course, before it is completely diluted, the labels are 'retained', but it is not because these cells are quiescent stem cell population, as is normally inferred from the term 'label retaining cell'. I suggest to edit the text around here to simply convey the point that 'transient-labeling of nucleus by histoneH2B-GFP allowed to monitor these cells' proliferation status'. The following FUCCI data are also along the same time – they are not quiescent and cycling.

---

## [Author Response]

[…]

*Essential revisions:*

*Several points were raised that have to be addressed prior to publication. The reviewers agreed that they only involve relatively straightforward revision experiment(s) and text editing. Thus, we would like to invite you to submit a revised version. Requested revisions are outlined below.*

*1) Lineage tracing experiments: in order to show the multipotency, the clones should be induced at low frequency and ideally at single cell level, and then show that a single labeled HSC can expand and differentiate into different cells in this lineage.*

We do agree with the reviewer that it is more appropriate to induce clones at low frequency to establish the multipotent nature of the Notch expressing cells.

In this context, we want to draw the attention of the reviewer that we had already induced clones at single and double cell level and have shown that they do indeed expand and differentiate into different cell types including the progenitors. These results are illustrated in Figure 3F-G and discussed in the text (subsection “The Notch expressing cells in the first instar lymph gland are multi-potent stem cells”, fourth paragraph) in the revised manuscript.

In complementary, these potential HSCs could be genetically ablated and show no progenies arising from them. Along the same line, in Figure 5, the authors made a strong statement that those N-expressing cells are founder cells for those Dome-expressing cells. However, some of the later stage Dome-expressing cells also come from N-negative cells. It is OK that more than one founder cell exists. But in order to complement the tracing experiments, more convincing evidence would be ablation of those N-positive cells lead to significant decrease of Dome-cells, similar to the experiments in Figure 6. If these experiments are proven to be technically challenging, please elaborate on this point in the response letter.

This is a wonderful suggestion that would serve as a functional correlate of the results of our lineage tracing experiments.

We have now induced genetic ablation of Notch expressing cells. This was achieved by the activation of Reaper (rpr, pro-apoptotic gene) specifically in the HSCs by Notch-Gal4. To start with, this turned out to be a very challenging experiment. Since Notch regulates diverse early developmental events, upon driving *UAS-rpr* by *Notch-Gal4* we encountered early lethality. However, after several trials we were successful in narrowing down the activation window for reaper to four hours post emergence of larvae. This strategy yielded few individuals that were lethal at late third instar stage.

To ensure successful ablation of Notch expressing cells, *dome-MESO-LacZ* was brought inthe background of *Notch-Gal4*. On restricted activation of *UAS-rpr*, few individuals of every batch were analyzed at first instar stages to ensure that the lymph gland contains only *dome-MESO-LacZ* and no Notch expressing cells thereby confirming a successful ablation (please compare Figure 6A with Figure 6C). The siblings of the batch that had only *dome-MESO-LacZ* expressing cells in the first instar larval lymph gland and made to third instar stage were analyzed. We found that genetic ablation of Notch expressing cells causes a massive reduction in the lymph gland size at third instar (please compare Figure 6B with Figure 6D). Quantitative analyses revealed that an average 5-fold reduction in the total size of the lymph gland (Figure 6E). Moreover, the ratio of area of progenitor cell population versus the total lymph gland area got reduced by more than 2 fold (Figure 6F) indicating that by killing the Notch expressing founder cells, the reserve population of progenitors arising from them were also eliminated. This in turn resulted in a massive reduction in the total lymph gland size that now houses the clonal expansion of those Dome expressing (but not Notch) progenitors that coexisted with the Notch positive HSCs.

We sincerely thank the reviewer for his/her suggestion. We can conclude that the *Notch* positive cells are the multi-potent founder cells or the HSCs. This experiment has indeed enriched and strengthened our manuscript to a great extent.

*2) The overall characterization of co-expression vs. mutually exclusive patterns in Figure 1 should be done with co-labeling. If the figure only has one staining, it is hard to make such a claim, such as in Figure 1C, 1E-G, H, etc.*

In our manuscript, panels in Figure 1 and Figure 1—figure supplement 1 were so arranged that the later complement the former.

For instance:

A) Figure 1C (demonstrating dot expression in the HSCs) is supported by Figure 1—figure supplement 1D where we show dot expression with Antennapedia marking the PSC/niche.

B) Figure 1D and Figure 1—figure supplement 1E provide two independent examples of first larval instar lymph gland showing the expression of dot with dome-MesoLacZ. These co-expression patterns demonstrate that the cells with high levels of Dot expression and those with dome-MesoLacZ expression are mutually exclusive.

C) Although only Notch expression is shown in Figure 1E, the Figure 1—figure supplement 1G shows both Notch and Antp expression demonstrating the absence of Notch from the niche cells.

Following the reviewer’s suggestion, we have now included Figure 1E’ that has Notch and dome double staining that clearly shows that they are mutually exclusive cell types.

D) Likewise Figure 1F demonstrates the expression of Su(H)lacZ (reporter of Notch activity) in the HSCs and Figure 1—figure supplement 1H-H**"** show that the Su(H)lacZ cells are also positive for Notch-GFP expression.

*3) They state that HSC expresses STAT: this is somewhat puzzling given that the progenitor cells are Dome-positive: the upstream receptor for STAT. What is the relationship between Dome+ cells and STAT+ cells? (if they are distinct cell types as inferred by authors, then Dome+ cells are not transducing STAT signaling? And how STAT+ cells can become STAT+ without expressing Dome?)*

We believe that the STAT signaling in larval lymph gland does not follow the canonical pathway. Several instances can be cited in this context.

For example, it has been demonstrated that the PSC/niche cells express STAT but they never express Dome. Interestingly, lineage tracing of dome population in developing larval lymph gland never marked PSC/niche cells (Jung et al., 2005).

More recently, STAT pathway has been implicated in maintaining the quiescence state of hematopoietic progenitors (Mondal et al., 2011). Interestingly, in this study also STAT signaling is elicited in the cortical cone (CZ) of the lymph gland that never expresses Dome.

*4) The authors report that the Notch expressing cells in the earlier time points are bigger in size, than the neighboring cells. And these cells diminish over time. However, the data presented to support this is not convincing. The Notch expressing cells in Figure 1M look much bigger than the ones in Figure 1N (1M scale bar is missing). Do these cells diminish in size over time? Are they bigger in 8h time point when compared to 13h. A quantitative analysis of the Notch expressing cell size over time should address this concern.*

Thanks for drawing our attention. Since the size of the lymph gland at 18h is relatively bigger than that of 13hrs, the 18h lymph gland was photographed in a different magnification. The scale bars in both the figures reflect this difference. We are extremely sorry for not including the scale bar for Figure 1M in the previous version that lead to this confusion!

Following the suggestion of the reviewer, we have done a quantitative analysis of the cell size of Notch expressing cells over time and found that their size remains more or less unaltered.

As evident from the graph shown in Author response image 1, we did not observe any significant change in the size of Notch expressing cells at 13h (p=0.842025521, n=14) or 18h (p=0.35903873, n=14) AEH with respect to those at 8h.

*5) Dome-meso-lacZ positive cells in Figure 1B" also have different level of lacZ, any explanation for that?*

We speculate that this variation is an outcome of differential accumulation of β-galactosidase in these cells.

*Also, it seems that Dome-meso-lacZ and dome-Gal4: UAS-GFP do not have totally overlapping signal in Figure 1—figure supplement 1C-C", why?*

Dome-meso-LacZ enhancer is a complete mesodermal specific reporter of JAK/STAT activation that mimics the expression of endogenous *dome* (Hombria et al., 2005). Although in third instar it co-localizes with Dome Gal4 UASGFP expression, but our study shows that it can sense JAK/STAT activity much earlier than the *Dome-Gal4* construct. We believe this may be due to its specificity as a mesodermal reporter. Moreover, the delay can also be attributed to the synthesis of Gal4 and subsequent activation of GFP. This becomes evident at the point of initiation of a gene expression where an enhancer might respond earlier than a Gal4 based reporter.

*Additional points:*

*Other comments have been raised regarding the clarity of the manuscript. Although these are not about the scientific contents of the manuscript, we recommend that these points are addressed prior to publication.*

*1) The manuscript can be improved by editing. Please try to increase the clarity of the manuscript to appeal to a broader readership of eLife.*

Following the suggestion of the reviewer we have tried our best to improve the manuscript and are open to any suggestion from the editors to make it more lucid.

*2) The figures and figure description need work, as there are some panels that are hard to decipher. Examples are:*

The figure panels are not necessarily in temporal order e.g. Figure 6 B-C.

*Also the Figure 4A-B' lettering is not aligned.*

Our apologies for this kind of error! We have taken care of these issues in the revised version.

*Figure 1Q is not necessary and can be mentioned in the text?*

As suggested, we have removed the figure and mentioned the observation in the text (subsection “The novel cell population is transient and expresses several molecular markers”, fourth paragraph).

*Figure 1B and Figure 1—figure supplement 1A-A' show the same data and should not be repeated.*

In the revised version we have removed Figure 1—figure supplement 1A-A'.

*The scale bars between Figure 2—figure supplement 2A and Figure 2—figure supplement 2B are not the same. But they seem to be of the same magnification?*

We are extremely sorry for this error. Thanks for pointing it out. They are of indeed of same magnification. We have corrected the error in the revised version.

*In Figure 6C a co-stain of LaminC with Dot/Antp would better ascertain their identity as PSC niche cells.*

We have included a costaining of Lamin C and Antp for 8h and 18h after egg hatching in the revised version. Current Panel is Figure 7B-C''.

*Figure 6D-D" and Figure 6—figure supplement 1C-C" are repetition of the same data, with the latter not being mentioned in the text.*

We have now mentioned this in the revised manuscript (subsection “Dpp signal from the PSC maintains the self-renewal of HSCs”, fourth paragraph).

*3) A little more information on Odd expression will help a broad audience understand this better.*

We have now included the required information in the revised version (subsection “Notch expressing multi-potent cells are the founder cell for progenitors of the larval lymph gland”, second paragraph).

*4) Does DPP affect size of the niche itself? They should monitor the niche size to see if the effects are direct or indirect. Also, how will Dpp/TKV overexpression effect the HSC population?*

Thanks for raising this point. Performing the analyses as suggested by the reviewer indeed reveals the direct effect of Dpp on the maintenance of HSCs.

Previous studies have demonstrated that *dpp* loss from the niche causes increment in niche cell numbers (Pennetier et al. 2012) at third instar stage of development. Similarly, when we analyzed third instar larval lymph glands, we also observed that the niche number gets affected in dpp heteroallelic condition. However, in this current study, we effectively demonstrate that loss of dpp, specifically in the first 18h AEH, affects the number of HSCs without having any obvious effect on the number of niche cells (Figure 7—figure supplement 1M).These results clearly establish that in the first instar larvae Dpp signaling from the niche (PSC) is essential for the sustenance of the HSCs. We have now included these results in the revised manuscript (subsection “Dpp signal from the PSC maintains the self-renewal of HSCs”, last paragraph).

We faced technical difficulties in overexpressing *dpp* within the niche as it resulted in larval lethality. This prevented us to assay the effect of dpp overexpression on HSC population. Even a pulse of dpp activation during HSC lifetime (0-18h AEH) resulted in early second instar lethality.

*5) Notch staining should be included in Figure 6E to show the range of niche-to-stem cell signaling in this system. Are the "bigger" high-level Notch expressing cells closer to the PSC?*

Two examples are already included in the panel: current Figure 7F and 7J.

Following reviewers suggestion, we analyzed about 50 first instar lymph gland from Enhancer of split. We did not find samples which endorsed that bigger high level Notch expressing cells are in close proximity to the PSC.

*6) The dual FUCCI system is easier to interpret in distinguishing between the cell cycle phases than the Geminin FUCCI reporter. The supplemental data that shows the dual reporter could be moved to the main figure.*

We have now rearranged the figures according to the reviewer’s suggestions in the revised manuscript.

*Also in Figure 2—figure supplement 1F, the authors distinguish between G2 and M phase cells using Dual FUCCI system. This seems inappropriate, as both phases would have a "yellow" nuclear stain. They can address this by staining for PH3 in the dual FUCCI system. The graph distinguishes these two cell cycle phases but the reporter does not. Did they do a PH3 in this system as well? It is not shown.*

Thanks for bringing it forth. We had actually done PH3 labeling on dual FUCCI which is reflected in the graph. We have now included three representative pictures of three different time points in the revised version (Figure 2K-M).

*7) Figure 2: the term "label retaining cell" is confusing. What the data show in this figure is actually histone label is simply diluted as N+ cells undergo division. Of course, before it is completely diluted, the labels are 'retained', but it is not because these cells are quiescent stem cell population, as is normally inferred from the term 'label retaining cell'. I suggest to edit the text around here to simply convey the point that 'transient-labeling of nucleus by histoneH2B-GFP allowed to monitor these cells' proliferation status'. The following FUCCI data are also along the same time – they are not quiescent and cycling.*

We have taken care to resolve this confusion. The term "label retaining cell" has been removed and we have followed the suggestion of reviewer while interpreting the results (subsection “Notch expressing cells in the first instar lymph gland undergo asynchronous 188 division from 13 AEH”, first two paragraphs).